# Pollen assemblages and distribution characteristics in surface sediments of karst caves on the Guizhou Plateau, southwestern China

Liang Tang[1], Ge Liu[2]*, Chong Xu[1], Qian Wang[1], Songtao Li[1], Yunlong Fan[1], Xiaoshuang Zhao[3]

**1** Guizhou Provincial Key Laboratory of Geographic State Monitoring of Watershed, School of Geography and Resources, Guizhou Education University, Guiyang, Guizhou, China, **2** State Key Laboratory of Black Soils Conservation and Utilization, Northeast Institute of Geography and Agroecology, Chinese Academy of Sciences, Changchun, Changchun, China, **3** State Key Laboratory of Estuarine and Coastal Research, East China Normal University, Shanghai, China

* liuge@iga.ac.cn

## Abstract

Cave sediments commonly contain crucial sedimentary evidence of past environmental change and human activity. Fossil pollen sequences within these deposits hold great potential for reconstructing paleoenvironmental changes and human–environment relationships; however, the extent to which cave pollen faithfully reflects the external environment remains controversial, especially in complex cave systems. This study conducted pollen analysis in surface sediment samples from two karst caves with complex geometry on the Guizhou Plateau. The pollen assemblages in surface sediment within 5–15 m of the entrance of complex caves with multiple entrances and passages or a single entrance and multiple chambers were highly similar to those of external surface soil/fresh moss samples and they exhibited strong correlations, indicating a good representation of external vegetation. Additionally, the high pollen concentration in the area made it an ideal sampling area for pollen analysis. In the middle-to-rear parts of the caves, although improved ventilation enhanced the representation of pollen assemblages for external vegetation, pollen concentrations were significantly lower than those near the entrance. This necessitates careful selection of samples for pollen analysis from cave sediments. Moreover, humid cave environments, animal transportation, and plant growth within the cave may lead to spatial heterogeneity in pollen assemblages, thereby affecting their representativeness of the external environment. The present study provides theoretical references for understanding the relationship between cave pollen assemblages and the external environment and offers important evidence for future archaeological and paleoenvironmental reconstruction studies in this region using cave pollen.

**Data availability statement:** All relevant data are within the manuscript and its Supporting Information files.

**Funding:** This research was jointly supported by several funding sources. The Guizhou Provincial Science and Technology Program (Basic Research on Guizhou Science and Technology-ZK [2022] General 333 and ZK [2022] General 336) fully funded study design and sample analysis and partially supported publication costs. The National Natural Science Foundation of China (42061001) provided the field sample collection. The Youth Foundation Project of Guizhou Province (Basic Research on Guizhou Science and Technology-QN [2025] 439) and the Research and Innovation Platform (Team) Support Program of Guizhou Education University (X2024019) jointly covered English language editing and part of the publication costs.

**Competing interests:** The authors have declared that no competing interests exist.

## 1 Introduction

In arid-semiarid areas and karst regions with well-developed rock fissures, where continuous terrestrial deposits (e.g., lacustrine and peat deposits) are scarce, cave sediments are regarded as one of the key archives of paleoenvironmental information [1–5]. Fossil pollen sequences preserved in these sediments hold considerable potential for reconstructing Quaternary paleovegetation and paleoclimate changes [6–13]. However, compared to open depositional systems like lakes and bogs, the semi-enclosed nature of caves introduces additional complexity in pollen sources, transport, deposition, and preservation processes [5,14]. This complexity leads to uncertainties in whether fossil pollen preserved in cave sediments faithfully reflects contemporaneous environmental information beyond the cave [5,14–18]. To reduce these uncertainties, research into modern cave pollen processes is indispensable, as it provides the critical modern analogue for interpreting the taphonomy and environmental significance of such archives [14,19,20].

To clarify the representativeness and reliability of cave pollen in reflecting the external environments, palynologists have conducted studies on the relationship between modern cave pollen assemblages and external pollen rain and vegetation composition [16,17,19,21]. For example, Burney and Burney used pollen traps to compare modern pollen spectra inside three caves in New York State over two years with pollen rain outside the caves [21]. They found a high similarity between the cave pollen spectra and the external pollen rain, which initially confirmed that cave pollen can reflect external environmental characteristics. Navarro et al., de Porras et al., and Fiacconi and Hunt analyzed the relationship between pollen assemblages in cave surface sediments and those in external surface sediments, as well as their correlation with average plant cover [16,17,19]. They found that modern cave pollen assemblages and their taphonomic processes are influenced by multiple factors, including the cave geometry, the type and activity of pollen transport agents, the mode and intensity of bioturbation, the composition and distribution of local vegetation, and the dryness of the cave. However, current research is mostly focused on simple caves with a single entrance and a single passage, such as sac-like or elongated caves [17,19,20]. Research on more complex caves with multiple entrances, passages, and chambers remains very limited, which greatly constrains our understanding of the relationship between cave pollen and external environment and also affects the expansion of cave pollen applications in paleoenvironmental reconstruction under complex environmental conditions.

The Guizhou Plateau in southwestern China has a highly developed and diverse karst topography [22], including more than 10,000 karst caves [23]. The diversity and complexity of these caves offer abundant material for systematic studies of modern cave pollen, which can help reveal the differences in pollen deposition processes among caves of different geometry and their complex relationships with external environments. Additionally, archaeological excavations have shown that these caves are important shelters for early humans (e.g., Zhaoguo Cave in Gui'an, Chuandong Cave in Puding, and Laoya Cave in Bijie), with archaeological layers extending over tens of thousands of years [24–28]. Fossil pollen preserved in these layers provide

valuable clues for understanding past vegetation and climatic conditions during the time of early human habitation, which is significant for attaining an in-depth understanding of the early human–environment interactions. Accurate reconstruction of the paleoenvironmental information preserved in the fossil pollen sequences from these archaeological layers relies on a comprehensive understanding of modern pollen deposition processes in cave systems. However, research on the relationship between modern complex cave pollen and external pollen rain in the Guizhou Plateau remains very limited [14], which not only restricts our ability to interpret the fossil pollen records in archaeological layers accurately but also affects paleoenvironmental reconstruction and archaeological research using cave pollen. Therefore, conducting research on the relationship between modern complex cave pollen and external pollen rain on this region is crucial. This research will also help reveal commonalities and differences in pollen deposition processes among caves of different geometry and expand the application of cave pollen in paleoenvironmental reconstruction and archaeological research.

In summary, this study selected two complex karst caves with different geometry on the Guizhou Plateau to conduct surface sediment pollen analysis. The goal of the study was to reveal the indicative role of complex cave surface pollen assemblages in reflecting external vegetation and to explore the potential factors influencing cave pollen taphonomic processes. The findings of this research could provide modern process references for future archaeological and paleoenvironmental studies using cave fossil pollen in this region.

## 2 Description of the caves

The Yinhegong Cave (also known as the dry section of Longtan Cave; 26°26′38″ N, 106°34′50″ E, 1190 m a.s.l.), located within the Tianhetan Scenic Area in Guiyang City, Guizhou Province (Figs 1A and 1B), is a natural karst cave that is a show cave. The cave is a complex system with four entrances (East, South, North upper, and North lower entrances, denoted as E1, E2, E3, and E4 in Fig 2A, respectively) and three levels with a total length of approximately 2000 m. The E3 entrance is about 8 m wide, 4.5 m high, and has a nearly elliptical shape. The intersection near the inner part of the two northern entrances (E3 and E4) (Fig 2A) and the deeper sections of the cave has active karst dripwater. The terrain gradually rises from the intersection to the E3 entrance, and one side of the cave floor is covered with loose cave sediments and rock debris, which is occasionally interspersed with rodent feces. Bird activity is occasionally observed at the cave entrance and a small number of bats roost inside. This section of the cave is usually closed, with very few visitors. Comprehensive field investigations revealed that the surrounding vegetation outside the cave had high coverage and was dominated by secondary forests of *Pinus massoniana*, accompanied by other species, such as *Ligustrum lucidum*, *Broussonetia papyrifera*, *Platycarya strobilacea*, *Toxicodendron vernicifluum*, *Cinnamomum officinarum*, *Magnoliaceae*, and landscaped bamboo and *Osmanthus fragrans* [29].

Dongkou Cave (26°24′11″ N, 106°28′57″ E, 1236 m a.s.l.), located in Machang Town, Guiyang City, approximately 10 km southwest of the Yinhegong Cave (Figs 1A and 1B), is developed at the foot of a mountain and features a complex structure with a single entrance, single passage, and multiple chambers (Fig 2B). The entrance is about 5 m wide, 3 m high, has a semi-elliptical shape, and is approximately 2 m above the external ground level. The cave floor is nearly level, and the passage is elongated with significant variations in width toward the interior. The rear of the cave contains two elliptical chambers (Chamber I and Chamber II), which are connected by a tubular passage measuring about 5 m long, 0.6 m wide, and 1.5 m high (Fig 2B). The distance from the entrance to the bottom of Chamber I is about 50 m, and there are multiple piles of bat guano and corn stalks on the floor. The northwest wall of Chamber II has abundant karst dripwater. Bats roost on the ceiling of this chamber, and a large amount of moist yellow clay covers the floor. There are also partially burned tree trunks, charcoal, and animal bones. Field surveys revealed that the upper and middle parts of the mountain, where the cave is located, are dominated by *P. strobilacea*, whereas the foothills and surrounding mountains are primarily covered with secondary *P. massoniana*, mixed with species such as *Carpinus pubescens*, *Itea yunnanensis*, *Quercus fabri*, *Quercus acutissima*, and *Trachycarpus fortunei* [29]. The surrounding depressions have been cultivated into cropland for corn and paddy fields.

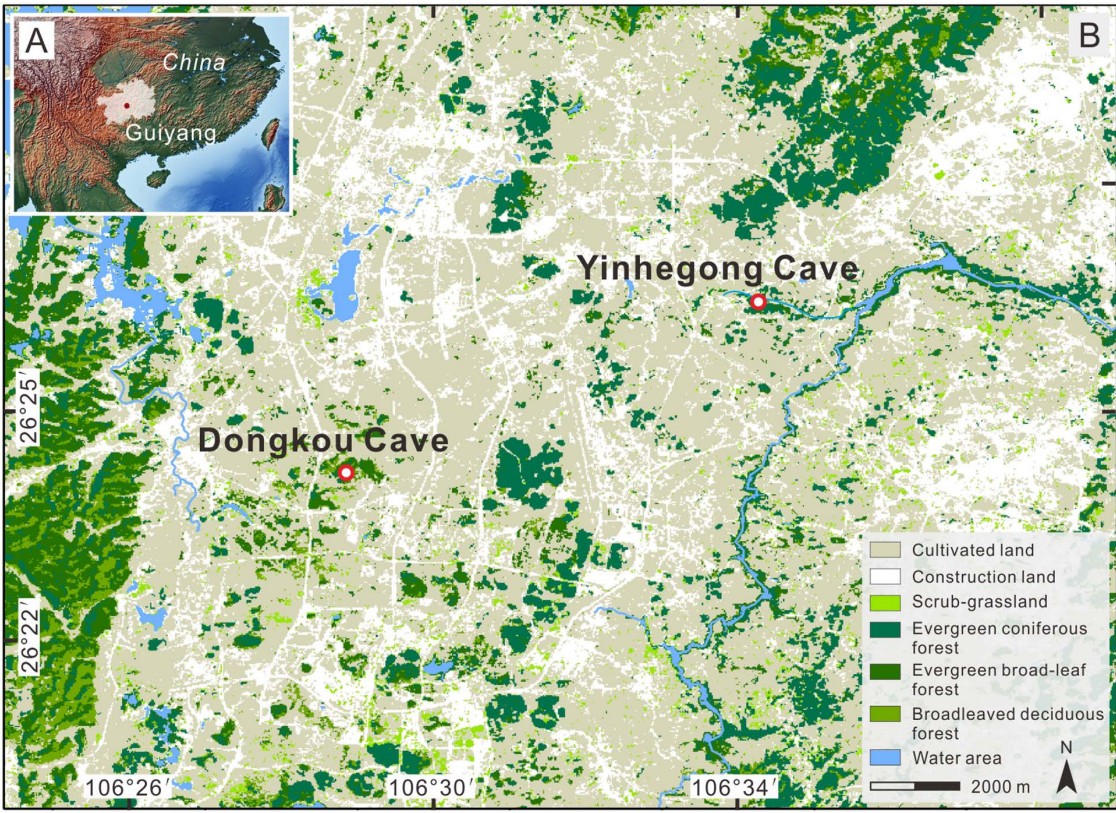

**Fig 1. Map of the study area.** (A) Location of Guiyang City, Guizhou Province, China. Map was provided by Natural Earth (http://www.naturalearthdata.com/), (B) Land cover around the Yinhegong and Dongkou caves and the location of the studied caves. The land cover map was generated with ArcGis 10.1 software (www.esrichina.com.cn) using the GLC_FCS30-2015 global land-cover products [30].

## 3 Materials and methods

### 3.1 Sampling

In October 2021, field investigations and sample collections were undertaken at Yinhegong Cave and Dongkou Cave. To minimize disruption to the cave surface sediments, sampling was conducted progressively from the entrance inward in each cave along a linear transect. Surface sediment samples measuring 10 cm × 10 cm in area and 1 cm in thickness were collected, placed into self-sealing bags, labeled, and brought to the laboratory for pretreatment. At Yinhegong Cave, samples were collected at 5 m, 10 m, 15 m, 25 m, and 30 m from the E3 entrance inward to the intersection of the E3 and E4 entrances (no sediment was present at 20 m due to exposed bedrock) and were sequentially numbered Y05, Y10, Y15, Y25, and Y30 (Fig 2A). At Dongkou Cave, nine samples were collected at 5-m intervals starting from 5 m inside the cave and labeled in order as D05, D10, D15, D20, D25, D30, D35, D40, and D45 (Fig 2B). Additionally, one sample was collected from the center of each of the two chambers labeled as Dc1 and Dc2 (Fig 2B).

To compare the relationship between cave surface pollen assemblages and external vegetation, three surface soil and fresh moss samples were collected in open areas outside the caves to represent the composition of the external vegetation [7,31]. A fresh moss sample, labeled Ym, was collected approximately 50 m northeast of the E3 entrance of Yinhegong Cave. In front of Dongkou Cave, a surface soil sample (Dr) was obtained 10 m from the entrance on a gentle slope, while one fresh moss sample (Dm) was collected from the slope over the entrance (Fig 2B).

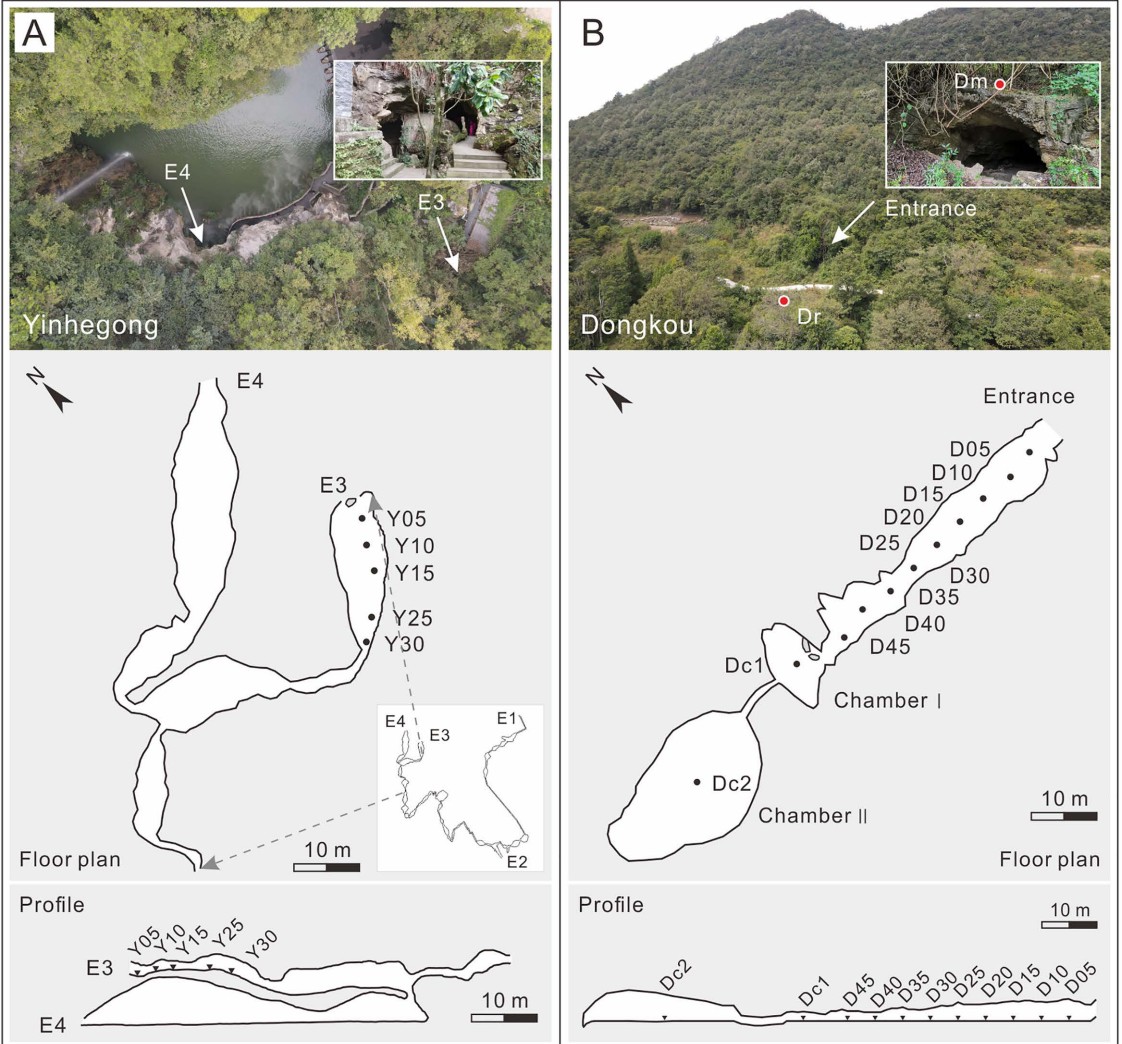

**Fig 2. Photographs, plan and sampling locations of Yinhegong (A) and Dongkou (B) caves (black dots indicate surface sediment sampling sites inside the cave, and red dots represent surface soil/fresh moss sampling sites outside the cave.** The fresh moss sampling site Ym outside Yinhegong Cave is not within the map area. In Fig 2A, E1, E2, E3, and E4 denote the four entrances, respectively; photography was performed by *Liang Tang* in October 2021 using a DJI drone).

### 3.2 Pollen analysis

A total of 19 samples were subjected to pollen pretreatment using the standard hydrofluoric acid (HF) method [32]. The procedures are as follows: approximately 8–15 mL of each dried sample was weighed and transferred into a 500-mL plastic beaker. To remove carbonates, 50 mL of 15% hydrochloric acid (HCl) solution was added to each sample, followed by repeated rinsing with distilled water until neutral. Subsequently, 50 mL of 40% HF solution was introduced to dissolve silicates, after which the samples were washed thoroughly again with distilled water to neutrality. The pollen residue was concentrated using a 7-μm nylon monofilament mesh in an ultrasonic bath to remove the tiny particles and then mounted in glycerin jelly for microscopic analysis. For quantitative pollen analysis, one tablet of *Lycopodium* spore (27,560 grains per tablet) was added to each sample as an exotic standard for calculating pollen concentration using the following formula:

$$Pollen\ concentration\ (grains/g) = \frac{L \times n}{l \times m}$$

<div align="right">(1)</div>

where *L* represents the number of *Lycopodium* spores added (27,560 grains), *l* denotes the number of *Lycopodium* spores counted, *n* is the pollen grains identified, and *m* refers to the sample weight (g or mL).

Pollen identification was performed using a Nikon ECLIPSE *Si* optical microscope (Nikon, Tokyo, Japan) at a magnification of ×400. Photographs of the main pollen types were taken and compiled into plates. For each sample, more than 200 terrestrial pollen grains (including both arboreal and herbaceous) were counted, and pollen percentages were calculated based on the sum of terrestrial pollen grains. Pollen diagrams were constructed using Tilia 2.6.1 software (https://www.neotomadb.org/apps/tilia). For pollen morphological identification, references were made to the following published works: *Pollen Flora of China* [33], *Angiosperm Pollen Flora of Tropic and Subtropic China* [34], *Sporae Pteridophytorum Sinicorum* [35], and *Atlas of Quaternary Pollen and Spores in China* [36].

To quantify the relationships between pollen assemblages in surface samples from inside and outside the cave further, Pearson correlation analysis was performed using OriginPro 2024 software (OriginLab Corp., Northampton, MA, USA).

## 4 Results

A total of 3,949 pollen grains were identified from 19 samples collected from the two caves, and these were classified into 69 families and genera. Specifically, there were 36 types of arboreal pollen, 18 types of herbaceous pollen, 12 types of fern spores, and 3 types of freshwater algae (Table 1 and Fig 3). In addition, a high abundance of fungal spores was recorded. However, only the number of fungal spores was counted in this study, and no detailed classification was made.

### 4.1 Yinhegong cave

Inside Yinhegong Cave, the surface sediment samples were dominated by arboreal pollen, with an average content of 94.7%, while herbaceous pollen accounted for 3.8% (Fig 4). The main arboreal taxa included *Pinus* (51.6%), *Carpinus* (2.2%), *Quercus* (1.9%), and *Betula* (1.7%), with a high abundance of *Pinus* bladder (30.0%). The herbaceous pollen was primarily composed of Asteraceae (1.2%), *Artemisia* (1.1%), and Cyperaceae (0.8%), with minor amounts of Ranunculaceae (0.2%), Brassicaceae (0.2%), Chenopodiaceae (0.2%), and Poaceae (<0.1%). The percentage of *Pinus* decreased twice from the entrance inward, while its bladder showed an inverse trend (Fig 4A). The percentages of *Carpinus*, *Betula*, *Quercus*, Asteraceae, and *Artemisia* increased toward the rear of the cave. Fern spores were commonly represented by undetermined Monoletes (40.1%), Gleicheniaceae (2.8%) and *Cibotium* (2.5%), with their highest percentages occurring in both the entrance and rear sections. Fungal spores were abundant and increased toward the rear of the cave. *Concentricystes* occurred only in the innermost sample Y30. Pollen concentration decreased toward the rear of the cave, with significant differences among samples. The highest concentration occurred in the entrance sample Y05 (353,870 grains/g), followed by Y15 (31,622 grains/g) and Y25 (21,335 grains/g), while the lowest was in the rear sample Y30 (6,634 grains/g) (Fig 4B).

The fresh moss sample outside Yinhegong cave had an arboreal percentage of 89.5%, with *Pinus* accounting for 73.4%, *Betula* (4.9%), *Carpinus* (3.3%), Oleaceae (1.8%), *Quercus* (1.8%), and *Pterocarya* (1.0%). The percentage of *Pinus* bladder was only 0.5% (Fig 4A). Herbaceous pollen was dominated by *Artemisia* (5.1%), followed by Poaceae <35 µm (2.8%), with minor amounts of Brassicaceae (0.8%), Ranunculaceae (0.5%), and *Zea mays* (0.3%). Ferns were primarily represented by Gleicheniaceae (1.0%), and the fungal spore content was lower than that inside the cave at 11.0%.

### 4.2 Dongkou cave

The surface sediment samples at Dongkou Cave had a higher percentage of arboreal pollen (55.1%) than of herbaceous pollen (43.1%) (Fig 5). Arboreal pollen was dominated by *Pinus* (23.3%), followed by Arecaceae (12.6%), *Carpinus*

**Table 1. Pollen and spore types in Yinhegong cave and dongkou cave.**

| Types | Yinhegong Cave | | Dongkou Cave | | Types | Yinhegong Cave | | Dongkou Cave | |
|---|---|---|---|---|---|---|---|---|---|
| | Exterior | Interior | Exterior | Interior | | Exterior | Interior | Exterior | Interior |
| *Pinus* | + | + | + | + | *Calophanoides* | | | + | + |
| Bladder | + | + | + | + | Caryophyllaceae | | | + | + |
| *Tsuga* | | | | + | Chenopodiaceae | | + | + | + |
| Taxodiaceae | + | | + | + | Cruciferae | + | + | + | + |
| Ephedraceae | | | | + | Compositae | | + | | |
| Anacardiaceae | + | | | | *Artemisia* | + | + | + | + |
| *Alnus* | | | + | + | *Aster* | | | + | + |
| *Betula* | + | + | + | + | *Taraxacum* | | | + | + |
| *Carpinus* | + | + | + | + | Cyperaceae | | + | + | + |
| *Corylus* | + | + | | + | *Veratrum* | | | | + |
| Celastraceae | | + | | + | Labiatae | | | | + |
| Ebenaceae | | + | | | Poaceae<35 μm | + | + | + | + |
| Euphorbiaceae | | + | + | + | Poaceae≥35 μm | | | + | + |
| *Phyllanthus* | | + | | + | *Zea mays* | + | | | + |
| *Sapium* | | + | | + | *Plantago* | | | | + |
| *Castanopsis/Lithocarpus* | | + | | + | Ranunculaceae | + | + | + | + |
| *Fagus* | + | | | | *Thalictrum* | | | + | + |
| *Quercus* | + | + | + | + | Polygonaceae | | | | + |
| Hamamelidaceae | + | | + | + | *Typha* | | | | + |
| *Liquidambar* | + | + | | + | Athyriaceae | | + | | |
| *Ilex* | | + | + | | *Cibotium* | | + | + | + |
| *Carya* | | | + | | Davalliaceae | | + | | + |
| *Juglans* | | | + | | *Humata* | | + | | |
| *Pterocarya* | + | + | | + | Gleicheniaceae | + | + | + | + |
| Leguminosae | | | | + | *Dicranopteris* | | | + | + |
| Magnoliaceae/Liliaceae | + | + | | | *Coniogramme* | | | | + |
| *Myrica* | | | + | + | Hymenophyllaceae | | | + | + |
| Moraceae | + | + | | | Lycopodiaceae | | | | + |
| Oleaceae | + | + | + | + | Selaginellaceae | | + | + | + |
| Palmae | | | + | + | *Osmunda* | | + | | |
| Rhamnaceae | | + | | | Polypodiaceae | | | + | + |
| Rutaceae | | | + | + | *Pteris* | | + | + | + |
| *Ulmus/Zelkova* | + | + | | + | *Concentricystes* | | + | + | + |
| *Celtis/Aphananthe* | | + | + | + | *Zygnema* | | | | + |
| *Salix* | | + | | + | *Spirogyra* | | | + | |
| Solanaceae | | | | + | fungal spore | + | + | + | + |
| *Vitis* | | + | | + | | | | | |

"+" indicates the presence this type of pollen in the sample.

(4.7%), *Quercus* (2.8%), *Betula* (2.7%), and Oleaceae (1.4%), with a small amount of *Pinus* bladder (4.6%) (Fig 5A). The herbaceous pollen was primarily composed of Poaceae <35 μm (9.5%) and Cyperaceae (9.1%), along with *Artemisia* (6.1%), *Taraxacum* (5.8%), *Aster* (5.3%), and Brassicaceae (3.3%). The percentages of *Pinus*, *Betula*, and *Artemisia* decreased toward the interior of the cave, while percentages of *Carpinus*, Oleaceae, and *Pinus* bladder increased.

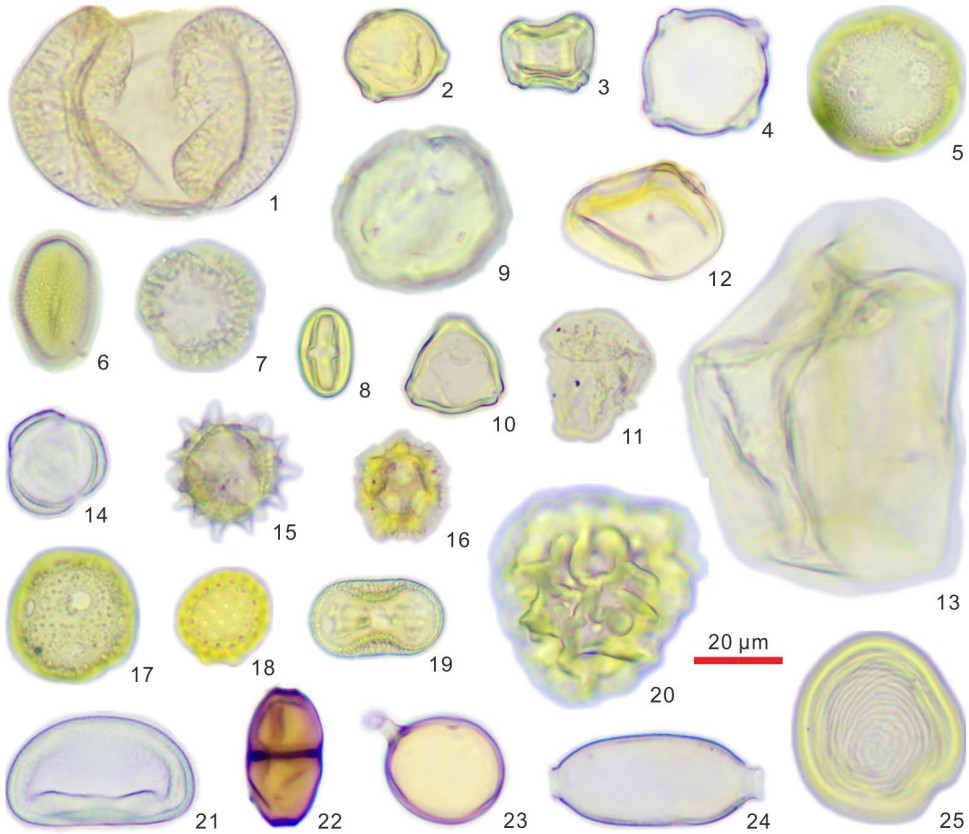

**Fig 3. Light microscope micrographs of typical pollen, spores, and non-pollen palynomorphs in the surface sediment of Yinhegong and Dongkou caves (1, *Pinus*; 2, *Betula*; 3, *Alnus*; 4, *Carpinus*; 5, *Liquidambar*; 6, Palmae; 7, Oleaceae; 8, *Castanopsis/Lithocarpus*; 9, *Juglans*; 10, *Myrica*; 11, Cyperaceae; 12, Poaceae <35 μm; 13, *Zea mays*; 14, *Artemisia*; 15, *Aster*; 16, *Taraxacum*; 17, Caryophyllaceae; 18, Chenopodiaceae; 19, *Calophanoides*; 20, *Cibotium*; 21, Polypodiaceae; 22, SAM-25; 23, *Glomus*; 24, *Trichuris* sp.; 25, *Concentricystes*).**

Arecaceae first increased and then decreased from the entrance inward, while Cyperaceae, fungal spores, and *Concentricystes* showed a significant increase toward the rear of the cave. Fern spores were mainly represented by Gleicheniaceae (5.0%) and Polypodiaceae (3.5%), which showed higher percentages in both the entrance and rear sections ([Fig 5B]). Pollen concentration was enriched at around 20 m from the entrance (samples D05 to D30, averaging 36,829 grains/g) and significantly decreased in the rear of the cave (samples D35 to Dc2, averaging 3,861 grains/g) ([Fig 5B]).

The surface soil and fresh moss samples outside the cave included an average percentage of arboreal pollen of 69.3%, dominated by *Pinus* (57.8%) along with *Betula* (1.7%), *Carpinus* (1.2%), Oleaceae (1.2%), Arecaceae (1.0%), and *Quercus* (0.7%) ([Fig 5A]). Herbaceous pollen accounted for 30.4% and consisted mainly of *Taraxacum* (10.9%), *Aster* (4.6%), Poaceae <35 μm (4.7%), *Artemisia* (3.2%), Cyperaceae (1.9%), and Brassicaceae (1.5%). Ferns were primarily represented by Polypodiaceae (9.0%) and Gleicheniaceae (6.7%). Fungal spores were less abundant than inside the cave, but the content of *Concentricystes* (2.8%) was comparable to the average content inside the cave. The average pollen concentration (30,139 grains/g) was similar to that at the cave entrance section (D05–D30) ([Fig 5B]).

### 4.3 Correlation analysis

The results of correlation analysis based on pollen percentage data ([S1] and [S2 Datasets]) indicate that the correlation coefficients (R) between the cave surface pollen assemblages and external pollen rain ranged from 0.18 to 0.99 ([Fig 6],

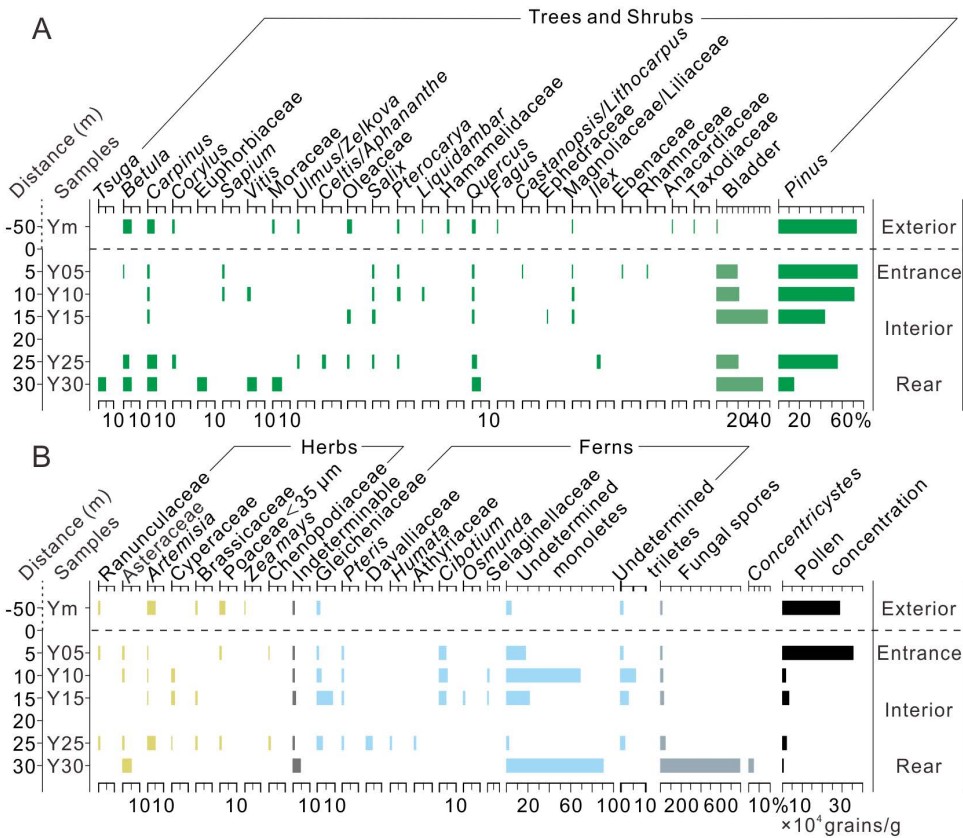

**Fig 4. Changes in the percentage and concentration of pollen at the Yinhegong Cave (S1 Dataset).**

S1 and S2 Tables). Specifically, at Yinhegong Cave, the R values ranged from 0.27 to 0.96; at Dongkou Cave, they ranged from 0.18 to 0.99 (Fig 6). Within Yinhegong Cave, the R values between the entrance samples (Y05 and Y10) and the rear sample (Y25) and the external moss samples exceeded 0.94, whereas the R values for samples Y15 and Y30 were less than 0.66 (Fig 6A). Similarly, within Dongkou Cave, samples near the entrance (D05–D15) and in the middle-to-rear sections (D35 and D40) exhibited high correlation coefficients with external samples (Dr and Dm, R > 0.70), while the remaining samples had low correlation coefficients (R < 0.65) (Fig 6B).

## 5 Discussion

### 5.1 Indicative role of cave surface pollen assemblages for external vegetation

Our pollen analysis of complex caves with various geometry indicated that in both the multi-entrance multi-passage Yinhegong Cave and the single-entrance multi-chamber Dongkou Cave with significant variations in passage width, the surface pollen assemblages within the 5–15 m section from the entrance showed a pattern similar to that of the external pollen rain, albeit with minor differences (Figs 4–6). The assemblages were dominated by *Pinus*, with minor contributions from *Betula*, *Carpinus*, *Quercus*, *Artemisia*, and Poaceae. The surface pollen assemblages corresponded to the typical vegetation features of the region, which were dominated by secondary forests of *P. massoniana* and open areas along the forest edges where *Artemisia* and Poaceae grew as weeds (Figs 4 and 5). The high correlation coefficients (0.99 > R > 0.77) between the cave surface pollen assemblages and the external samples further confirmed that the surface pollen

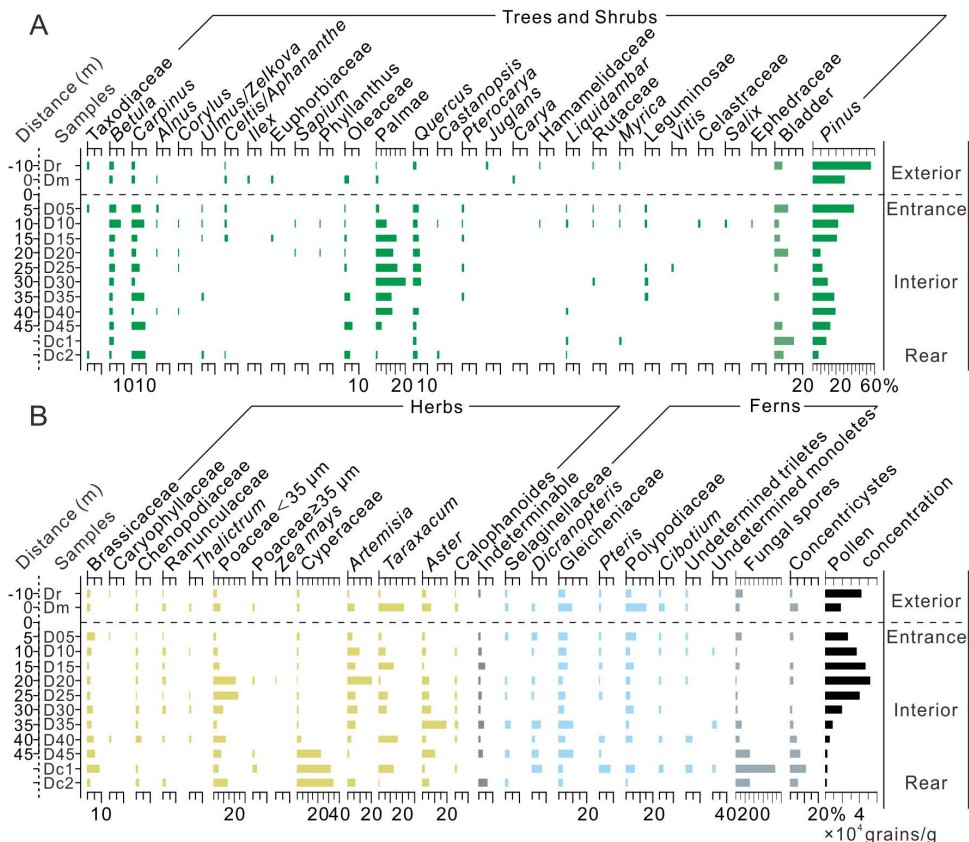

**Fig 5. Changes in the percentage and concentration of pollen at the Dongkou Cave (S1 Dataset).**

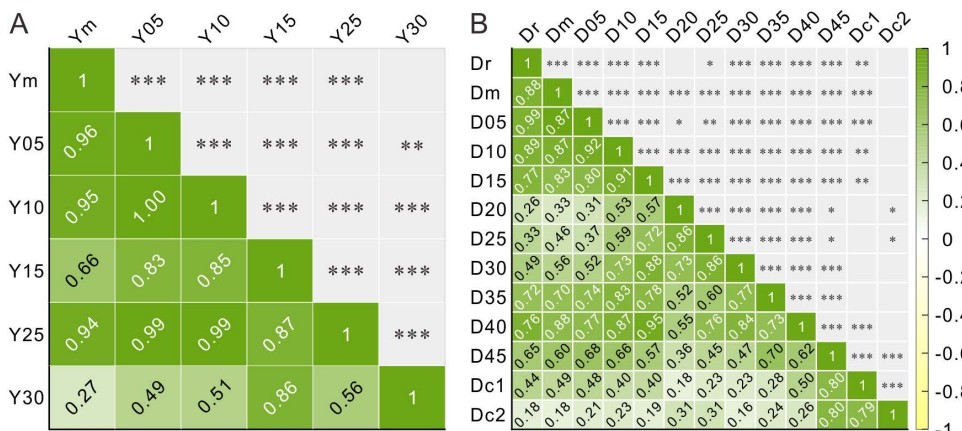

**Fig 6. Correlation coefficients for pollen spectra from the samples of Yinhegong Cave (A) and Dongkou Cave (B) (significance levels \*: P<0.05; \*\*: P<0.01; \*\*\*: P<0.001).**

assemblages within the 5–15 m range of the entrance reliably reflected the external pollen rain (Fig 6), indicating that the surface pollen assemblages in the section can effectively represent the external vegetation composition.

These findings are consistent with previous studies on caves in the karst regions of southwestern China, such as the typical long and narrow Zhongshan Cave and the Mohui Cave with a sack-shaped front and elongated rear section [14], as well as with studies of caves in other regions, including SLS203 Cave in Kurdish Iraq [20] and La Ultima and La Cocina Caves in the Patagonian Plateau of Argentina [19]. From the perspective of cave geometry, neither simple structures (e.g., sac-like or elongated) nor complex structures (e.g., multi-passage and multi-chamber) appeared to significantly influence the relationship between the surface pollen assemblages in the entrance section and the external vegetation. Therefore, we hypothesized that a good correspondence between the surface pollen assemblages in a certain range of the cave entrance and the external vegetation may be a universal pattern. However, the specific distance threshold for this correspondence needs to be defined further through statistical quantification of additional cave surface pollen analyses. To determine this distance threshold more precisely, future research should build on this foundation by expanding the sample size to include caves from diverse regions and with various geometries. This effort will provide a more universal theoretical basis for cave surface pollen studies.

However, this study also found that in complex caves, in addition to the entrance section, some samples from the middle-to-rear sections of the cave (Samples Y25, D35, and D40) showed good correspondence with the external pollen rain (Figs 4–6). This phenomenon has rarely been mentioned in previous studies. Based on the pollen assemblages and the geometry of the caves, we speculate that this may result from changes in the cave's internal structure that improve ventilation in the rear of the cave. Previous studies have also pointed out that better ventilation in a cave facilitates the transport and mixing of pollen [5,14,17]. In the complex, multi-entrance and multi-passage Yinhegong Cave, the sampled section connects to a lower side passage that opens to the exterior (Fig 2A). This connection may create a "flue" effect that enhances air circulation in the rear of the cave [37,38], thereby increasing the pollen influx from the north lower entrance (E4). The elevated pollen concentration and increased percentages of typical wind-pollinated taxa, such as *Pinus*, *Betula*, *Carpinus*, and *Quercus* in Samples Y15 and Y25, which are located approximately 20 m from the entrance (Fig 5), support this hypothesis. In contrast, the rear of Dongkou Cave does not have a side passage, but the percentages of wind-pollinated taxa, such as *Pinus*, *Carpinus*, and Poaceae also increase significantly. We observed that the passage width in Dongkou Cave rapidly expanded from about 3.4 m to 6 m at around 35 m from the entrance (Fig 2B). Therefore, we speculate that increased passage width may enhance the lateral circulation of air within the cave [17], promoting thorough mixing of pollen. However, the simultaneous increase in the percentages of entomophilous taxa (e.g., Oleaceae, Brassicaceae, *Taraxacum*, and *Aster*) in Samples D35–D45 (Fig 5) makes it difficult to rule out the contribution of animal-mediated transport to this pattern.

The studies mentioned above demonstrate that in complex caves the surface pollen assemblages from both the entrance and middle-to-rear sections with better ventilation can effectively represent the external vegetation. However, sufficient pollen concentration in samples is one of the essential conditions for ensuring the reliability of pollen analysis [14,39]. Given that the entrance area typically has high pollen concentrations, it is an ideal sampling area for pollen analysis. Although the pollen assemblages in the middle-to-rear sections of caves correspond well with the external pollen rain, these areas are not recommended as primary sampling locations because of their lower pollen concentrations.

## 5.2 Variability of cave surface pollen assemblages and its influencing factors

The surface pollen assemblages of Yinhegong Cave and Dongkou Cave both reflect a secondary forest dominated by *P. massoniana*, but percentage of arboreal taxa in Yinhegong Cave (94.7%) is significantly higher than that in Dongkou Cave (55.1%). This marked difference indicates distinct vegetation cover in the vicinity of the two caves. Yinhegong Cave, as a tourist attraction, has implemented vegetation restoration and conservation measures, primarily focusing on artificial cultivation of trees and shrubs, resulting in high forest cover (Figs 1B and 4). In contrast, Dongkou Cave is located near a

village where human activities, such as logging and slash-and-burn practices, occur periodically, leading to relatively lower forest cover (Figs 1B and 5). Within the same cave, the surface pollen assemblages also exhibit significant spatial heterogeneity, which may reflect the influence of factors such as environmental humidity, biological activity, and local vegetation—both outside or inside the cave—on the sources, transport, deposition, and preservation of cave pollen [5].

**5.2.1 Cave environmental humidity.** As mentioned previously, enhanced ventilation facilitates the transport of pollen into a cave [14,17,37]. According to this principle, the rear of Yinhegong Cave, which is connected to a side passage closer to another external entrance (E4) (Fig 2A), should exhibit higher pollen concentrations and a greater diversity of pollen types. However, the opposite occurs: the pollen concentration in Sample Y30 from the rear drops to the lowest level throughout the cave, and the number of pollen types (10) is reduced by approximately half compared to the preceding Sample Y25 (19) (Fig 4). Investigations of the cave environment revealed that the rear of the sampled section of Yinhegong Cave (beyond Sample Y30) was characterized by active karst dripwater and high humidity. Although the present study did not monitor the humidity levels of the cave, monitoring results from similar caves in the region (e.g., Mahuang Cave, Zhijin Cave) have shown a significant increase in cave environment humidity from the entrance to the rear [40,41]. Therefore, we speculate that the significant reduction in pollen concentration and diversity in sample Y30 results from pollen degradation caused by the humid cave environment in the rear. Davis noted that dry cave environments, such as those found in Bechan in the western United States, are conducive for pollen preservation, whereas humid conditions tend to promote microbial activity, which accelerates pollen decay and makes the environment unfavorable for pollen preservation [6]. Similar conclusions have been reported by Navarro Camacho and Navarro et al. in their investigations of five caves in southeastern Spain [7,16,42]. Our pollen analysis results support the above speculation. For example, the presence of *Concentricystes*, an indicator of humid environments [43–45], in Sample Y30 from Yinhegong Cave (Fig 4), along with the increased proportion of fungal spores reflecting microbial activity [16] and the increase in the percentage of indeterminable pollen grains due to decay and decomposition [16,17] (Fig 4), all suggest that a humid environment may have led to pollen degradation.

**5.2.2 Role of animals in transporting pollen.** Pollen transport by animals is also an important factor affecting pollen deposition in caves [5,21,46]. Research has shown that vertebrates, insects, and other animals (such as bats and bees) can introduce pollen into caves and accumulate it in their activity areas through behaviors, for instance by roosting and nesting within caves [46]. In this study, we observed that the percentage of pollen from the Asteraceae (such as *Artemisia*, *Aster*, and *Taraxacum*), Poaceae (<35 μm), and Brassicaceae in Dongkou Cave increased significantly at 20–25 m (Samples D20–D25) and 40 m (Samples D40–Dc2) inward from the entrance and that the percentage of Palmae pollen increased notably near the middle of the cave at around 30 m (Fig 5). This is likely related to animal transport. First, in terms of pollination mechanisms, these pollen grains are predominantly transported by animals. For example, Asteraceae and Brassicaceae are primarily dispersed by insects [16,47]. Although Poaceae is a type of anemophilous pollen, it can also be transported by insects [16]. Additionally, Palmae mainly relies on animal-mediated dispersal [48]. Second, there were large numbers of bats roosting on the ceilings of the two inner chambers and bat guano piled up on the floor near the middle of the cave at 20 m in Dongkou Cave. Previous studies have shown that bat guano is rich in pollen grains [42,49,50]; consequently, the intermittent increase in entomophilous pollen in the middle and rear sections of the cave may be attributable to bat activity.

**5.2.3 Growth of plants inside the cave.** This study also found that the percentage of fern spores in the entrance and middle-to-rear/rear sections of both Yinhegong Cave and Dongkou Cave was relatively high (Figs 4 and 5). Based on vegetation surveys outside the caves, we attribute the high content of fern spores at the entrance to the growth of ferns around the cave opening. In the pollen analysis of C7 and Dog Hole caves at Creswell Crags, Coles and Gilbertson noted that an increase in fern spores at the front of caves under humid climatic conditions reflects the growth of ferns at the cave entrance [37]. Similarly, Fiacconi and Hunt found in their study of SLS207 cave that the growth of ferns on the cave walls at the entrance corresponds to elevated spore content in the pollen spectra [17]. However, the phenomenon of increased

fern spore content in the middle-to-rear/rear sections of caves has rarely been mentioned in previous studies (Figs 4 and 5). Although the rear of Yinhegong Cave is connected to a side passage that enhances ventilation, Sample Y30 is located far from the intersection and entrances (Fig 2A). Additionally, fern spores are relatively large and have limited transport distances [51,52], making it unlikely that the source is external. In fact, within Yinhegong Cave, specifically in the deeper sections beyond Sample Y30, we observed lampenflora—a community of phototrophic organisms thriving under artificial light [53–55]. This community was dominated by germinating ferns colonizing the sediment-covered cave floor around the light fixtures. The ferns thrive under the combined conditions of moisture from cave dripwater and light provided by the artificial source [55]. Consequently, the high fern spore content detected in the samples from the rear of the cave likely originates from the lampenflora near the light sources. In Dongkou Cave, the elevated fern spore content in the rear corresponds with high values of insect-pollinated taxa, such as *Taraxacum*, *Aster*, and Brassicaceae (Fig 5), subtly suggesting that their transport might be mediated by animal activity.

Previous studies have also investigated the effects of cave dripwater and human activities on the transport and deposition of cave pollen [5,17,56]. Although both Yinhegong Cave and Dongkou Cave feature active karst dripwater in the rear, we did not perform pollen analysis on the dripwater or the corresponding floor sediments. Additionally, no pollen types introduced by seepage through cave roof fissures were identified in the pollen spectra [17]. Both caves have experienced varying degrees of human activity. For example, as a tourist cave, Yinhegong Cave is frequently visited by tourists entering through the northern lower entrance (E4) and moving inward to the cave's interior. Dongkou Cave shows evidence of human use of fire and other activities, but the pollen spectra do not exhibit signals of significant human disturbance [5]. Moreover, previous research has determined the spatial representation of cave pollen spectra (local vs. regional) through comparative analysis of cave surface pollen assemblages with external pollen rain and vegetation composition at varying spatial scales [7,19,20,37,42]. Case studies demonstrate distinct spatial patterns: surface pollen assemblages from Argentina and Kurdish Iraq primarily reflect local vegetation within a limited radius of tens to hundreds of meters [17,19]. Conversely, surface pollen records from England and Spain exhibit regional vegetation signals, although with significant spatial variation (5–20 km) [16,37]. In this study, while surface pollen assemblages from cave entrances and specific interior samples effectively reflect external vegetation characteristics, the precise spatial range represented by these assemblages remains unclear. This limitation stems from collection of external samples being confined to areas within tens of meters of the cave entrances. To address the limitation, future research should incorporate systematic surface sampling across broader spatial gradients, coupled with detailed vegetation surveys, to assess the spatial extent of vegetation signals preserved in cave surface pollen records quantitatively.

## 6 Conclusions

This study performed pollen analysis on 19 surface samples collected from two complex karst caves with distinct geometry on the Guizhou Plateau. In both the multi-entrance, multi-passage Yinhegong Cave and the single-entrance, multi-chamber Dongkou Cave with significant passage width variation, the pollen assemblage in the surface sediment within 5–15 m of the entrance exhibited strong correspondence with the external pollen rain, effectively reflecting the composition of the external vegetation. Improved ventilation in the rear of caves may enhance the representativeness of the surface pollen spectra for external vegetation in such parts of a cave. However, given the impact of pollen concentration on the accuracy of pollen analysis, the entrance area, which has higher pollen concentrations, is considered an ideal sampling section for cave pollen analysis. Both caves reflect the typical local vegetation dominated by secondary *P. massoniana* forests; however, the significant differences in percentages of arboreal pollen between the two caves indicate variations in average forest cover around the caves. The differences in surface pollen assemblages within the same cave can be attributed to multiple factors, including cave humidity, animal transportation, and plant growth inside the cave. The present study enhances our understanding of the relationship between cave pollen assemblages and the external environment and provides a useful foundation for further research.

## Supporting information

**S1 Table. Correlation coefficients for pollen spectra from the samples at Yinhegong Cave (significance levels *, P<0.05; **, P<0.01; ***, P<0.001).**
(DOCX)

**S2 Table. Correlation coefficients for pollen spectra from the samples at Dongkou Cave (significance levels *, P<0.05; **, P<0.01; ***, P<0.001).**
(DOCX)

**S1 Dataset. Pollen, spore, algae percentages and pollen concentration from Yinhegong Cave.**
(XLS)

**S2 Dataset. Pollen, spore, algae percentages and pollen concentration from Dongkou Cave.**
(XLS)

## Acknowledgments

We sincerely thank Dr. Ting Chen of the School of Geography and Tourism, Chongqing Normal University, for her valuable suggestions for improving the manuscript. We are also grateful to Professor Xinrong Liao for his help in field sampling.

## Author contributions

**Conceptualization:** Liang Tang, Ge Liu.

**Data curation:** Liang Tang.

**Formal analysis:** Liang Tang, Ge Liu, Chong Xu, Qian Wang, Songtao Li, Xiaoshuang Zhao.

**Funding acquisition:** Liang Tang.

**Investigation:** Liang Tang, Chong Xu, Qian Wang, Songtao Li, Yunlong Fan.

**Methodology:** Liang Tang, Yunlong Fan.

**Project administration:** Liang Tang, Ge Liu.

**Resources:** Ge Liu, Songtao Li.

**Software:** Liang Tang, Qian Wang, Xiaoshuang Zhao.

**Supervision:** Ge Liu.

**Validation:** Liang Tang, Ge Liu.

**Visualization:** Liang Tang, Chong Xu, Qian Wang, Xiaoshuang Zhao.

**Writing – original draft:** Liang Tang, Ge Liu, Chong Xu, Qian Wang, Songtao Li, Xiaoshuang Zhao.

**Writing – review & editing:** Liang Tang, Ge Liu, Yunlong Fan.

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
