## [Decision Letter · Decision Letter 0]

14 Mar 2025

Dear Dr. Tang,

Thank you for submitting your manuscript to PLOS ONE. After careful consideration, we feel that it has merit but does not fully meet PLOS ONE’s publication criteria as it currently stands. Therefore, we invite you to submit a revised version of the manuscript that addresses the points raised during the review process.

We look forward to receiving your revised manuscript.

Kind regards,

Tzen-Yuh Chiang

Academic Editor

PLOS ONE

Journal Requirements:

2. In your manuscript, please provide additional information regarding the specimens used in your study. Ensure that you have reported human remain specimen numbers and complete repository information, including museum name and geographic location.

For more information on PLOS ONE's requirements for paleontology and archeology research, see https://journals.plos.org/plosone/s/submission-guidelines#loc-paleontology-and-archaeology-research .

 “the Science and Technology Program of Guizhou Province (Basic Research on Guizhou Science and Technology-ZK [2022] General 333 and ZK [2022] General 336), the National Natural Science Foundation of China (42061001), the Scientific Research Project of Higher Education Institutions of Guizhou Provincial Department of Education (Youth Project: QJJ [2022] 256), the Research and Innovation Platform (Team) Support Program of Guizhou Education University (X2024019).”

4. We note that your Data Availability Statement is currently as follows: “All relevant data are within the manuscript and in Supporting Information files.”

5. We note that Figure 1 and 2 in your submission contain map/satellite images which may be copyrighted. All PLOS content is published under the Creative Commons Attribution License (CC BY 4.0), which means that the manuscript, images, and Supporting Information files will be freely available online, and any third party is permitted to access, download, copy, distribute, and use these materials in any way, even commercially, with proper attribution. For these reasons, we cannot publish previously copyrighted maps or satellite images created using proprietary data, such as Google software (Google Maps, Street View, and Earth). For more information, see our copyright guidelines: http://journals.plos.org/plosone/s/licenses-and-copyright.

 a. You may seek permission from the original copyright holder of Figure 1 and 2  to publish the content specifically under the CC BY 4.0 license. 

Reviewers' comments:

Reviewer's Responses to Questions

**Comments to the Author**

1. Is the manuscript technically sound, and do the data support the conclusions?

Reviewer #1: Yes

2. Has the statistical analysis been performed appropriately and rigorously?

Reviewer #1: I Don't Know

3. Have the authors made all data underlying the findings in their manuscript fully available?

Reviewer #1: No

4. Is the manuscript presented in an intelligible fashion and written in standard English?

Reviewer #1: Yes

Reviewer #1: The topic of the article is interesting as it provides insights into how pollen reaching cave environments reflects surface vegetation. In this sense, the study partially meets its objective, offers valuable information on the subject, and is, in my opinion, of interest for publication. That said, the manuscript requires substantial revisions and further refinement before it can be considered for publication in PLOS One. Below are my comments for evaluation, consideration, and/or discussion by the editor and the authors. If there are any disagreements, please do not hesitate to reply or discuss them.

Main comments:

*Title: I believe the title does not fully reflect the results of the study, as the pollen assemblages are assessed but not interpreted within an environmental context or in terms of their significance. I suggest rephrasing it to better align with the study's discussion and findings.

*Materials and methods

-Sampling: It is unclear whether the same volume of sediment was sampled at each sampling point, which is essential for making valid comparisons. While the thickness of the sediment is mentioned, the surface area sampled is not specified. If this information cannot be provided, it may be difficult to make quantitative comparisons between samples of different volumes.

-Pollen analysis: Similar to the sampling concerns mentioned above, it is unclear what volume of sediment was initially extracted to obtain the 10 g of dried sample for further analysis. Clarifying this information would improve consistency and comparability between samples. I believe the methodology used for pollen analysis should be described in greater detail, particularly highlighting the key steps. Providing a clear and structured explanation of sample preparation, extraction techniques, chemical treatments, and identification criteria would enhance the study’s transparency and reproducibility. Additionally, specifying any quantification methods applied, would further strengthen the methodological clarity.

*Results

-Table 1 should present the spore-pollen types for each cave separately, rather than in the current format. As it stands, it is difficult to determine which types are found in each cave. Given that the authors discuss both caves separately, as they should, I recommend restructuring the table to emphasise this distinction more clearly.

-Correlation analysis: It is unclear which dataset was used for the correlation analysis. To ensure reproducibility and reusability, I recommend providing the dataset as a supplementary file (the specific pollen counts being compared should be explicitly stated for a meaningful interpretation). Furthermore, the statistical significance of the correlation (p-value) is not reported. To ensure a complete statistical interpretation, both R and p-values should be provided.

*Discussions and Conclusions

-Section: Cave environmental humidity – The relative humidity measured at each sampling point should be included in your assessment. As it stands, this section is general and primarily based on common knowledge that cave humidity is typically high. However, you do not provide any specific humidity measurements or discuss its variation in your results. Including this data would strengthen the analysis and help contextualise its potential influence on pollen preservation and deposition.

-General comment: The authors discuss factors influencing forest cover around the caves and its impact on pollen counts; however, these aspects are only vaguely mentioned, despite their crucial role in shaping the observed patterns. Forest cover can be easily inferred from existing maps using Land Use/Land Cover Classification (GIS & Spatial Analysis), a relatively simple and time-efficient approach that would enhance the study's interpretation.

Alternatively, models estimating vegetation cover based on modern pollen deposition patterns in similar environments could provide valuable insights. Historical land cover data, aerial imagery, and satellite observations could also be used to compare present-day vegetation with past trends. Additionally, integrating environmental and climatic data, such as temperature, precipitation, and topographic features, would help contextualise vegetation distribution.

To improve the robustness of the analysis, these factors should be incorporated into multifactorial analyses, either as quantitative or qualitative variables, to better assess the relationship between surface and subsurface pollen assemblages and their environmental drivers. If none of these approaches are feasible, the authors should explicitly acknowledge this limitation and discuss its potential impact on their conclusions.

Minor comments and questions

*Abstract

- Lines 14-15: The rationale for selecting these two specific caves remains unclear within the context provided by the authors.

*Introduction

-Line 33: Providing a specific timespan would help clarify the temporal resolution of pollen analysis in reconstructing palaeovegetation or palaeoclimate.

-Line 48: Could you please elaborate on what specific subsequent research you or the referenced authors are referring to? Providing more context or examples would help clarify how this study contributes to or aligns with future research directions.

-Lines 74-76: The connection between the study’s aims and the archaeological records or past human occupation is unclear. As currently presented, these lines do not provide relevant context for the research focus and should be removed unless further justification is provided.

-Lines 94-98, 112-115: Relevant references should be included to support these statements. If these descriptions are based on the authors’ own evaluations, they should be moved to the Results section rather than remaining in the introduction.

*Results

-Line 202: The mention of surface sediment should be removed here, as it was already discussed in the previous paragraph. The current paragraph appears to focus on mosses, and keeping surface sediment in this context may cause confusion.

*Discussion

-Line 225: Please clarify what the consistent pattern refers to or is in relation to. The statement lacks clarity and should specify the context or comparison being made.

-Lines 249-251: These statements belong to the Results section and should be moved accordingly for better structure and clarity.

-Lines 274-276: On what grounds is this statement based? Please provide more specificity, supporting data, or a relevant reference to substantiate the claim.

-Lines 334-338: The phenomenon described, specific to caves modified for tourism or show caves, is known as "lampenflora". Please expand on this concept in your discussion and include relevant comparisons with existing literature that examines "lampenflora", its ecological implications, and how it relates to the findings of your study.

-Lines 351-357: It would be valuable to elaborate further on how vegetation from different distances can be represented in pollen assemblages found inside caves. Are there estimates on the spatial scale of vegetation input based on existing studies? Vegetation maps detailing areas at varying proximities to the caves are likely available, and incorporating such data would enhance the study by providing an estimate of the pollen source area. This would be particularly useful for researchers working in this field, helping them contextualise similar findings in different settings.

*Conclusions

-Line 367: It is unclear why the middle-to-rear parts of caves should be treated with caution. Could you clarify in relation to what? Are you referring to pollen deposition patterns, post-depositional processes, or other environmental factors affecting pollen preservation? Providing a more detailed explanation would help readers better understand the reasoning behind this statement.

**Do you want your identity to be public for this peer review?** For information about this choice, including consent withdrawal, please see our Privacy Policy

Reviewer #1: No

---

## [Author Response · Author response to Decision Letter 1]

28 Apr 2025

Reviewer #1: The topic of the article is interesting as it provides insights into how pollen reaching cave environments reflects surface vegetation. In this sense, the study partially meets its objective, offers valuable information on the subject, and is, in my opinion, of interest for publication. That said, the manuscript requires substantial revisions and further refinement before it can be considered for publication in PLOS One. Below are my comments for evaluation, consideration, and/or discussion by the editor and the authors. If there are any disagreements, please do not hesitate to reply or discuss them.

[Authors’ response] We would like to sincerely thank you for taking your valuable time to review our manuscript. All your comments and suggestions have been taken into account while revising our manuscript, and they have been very helpful for us to improve the quality of our manuscript.

Main comments:

1. Title: I believe the title does not fully reflect the results of the study, as the pollen assemblages are assessed but not interpreted within an environmental context or in terms of their significance. I suggest rephrasing it to better align with the study's discussion and findings.

[Authors’ response] We agree and your point is well taken. We have rephrasing it to: “Pollen assemblages and distribution characteristics in surface sediments of karst caves on the Guizhou Plateau, southwestern China” (Lines 1–2).

2. Materials and methods

-Sampling: It is unclear whether the same volume of sediment was sampled at each sampling point, which is essential for making valid comparisons. While the thickness of the sediment is mentioned, the surface area sampled is not specified. If this information cannot be provided, it may be difficult to make quantitative comparisons between samples of different volumes.

[Authors’ response] Sorry for the confusion. We have supplemented the information on the area of the surface sediment samples collected (10 cm×10 cm in area) in Lines 120–121 of the revised manuscript, in accordance with the actual situation of field sampling, to enhance the reproducibility of the sampling method.

3. Pollen analysis: Similar to the sampling concerns mentioned above, it is unclear what volume of sediment was initially extracted to obtain the 10 g of dried sample for further analysis. Clarifying this information would improve consistency and comparability between samples. I believe the methodology used for pollen analysis should be described in greater detail, particularly highlighting the key steps. Providing a clear and structured explanation of sample preparation, extraction techniques, chemical treatments, and identification criteria would enhance the study’s transparency and reproducibility. Additionally, specifying any quantification methods applied, would further strengthen the methodological clarity.

[Authors’ response] Thank you for your careful review. Yes, in fact, standards for sample preparation, extraction techniques, chemical treatments, and identification criteria have become very mature. We have revised this part to: approximately 8–15 mL of each dried sample was weighed. Please see line 141.

In addition, we have modified the Pollen analysis section in detail following your advice. Now, it changed to: The procedures are as follows: approximately 8–15 mL of each dried sample was weighed and transferred into a 500–mL plastic beaker. To remove carbonates, 50 mL of 15% hydrochloric acid (HCl) solution was added to each sample, followed by repeated rinsing with distilled water until neutral. Subsequently, 50 mL of 40% HF solution was introduced to dissolve silicates, after which the samples were again thoroughly washed again with distilled water to neutrality. The pollen residue was concentrated using a 7–µm nylon monofilament mesh in an ultrasonic bath to remove the tiny particles and then mounted in glycerin jelly for microscopic analysis. For quantitative pollen analysis, one tablet of Lycopodium spore (27,560 grains per tablet) was added to each sample as an exotic standard for calculating pollen concentration using the following formula:

(1)

Where L represents the number of Lycopodium spores added (27,560 grains), l denotes the number of Lycopodium spores counted, n is the pollen grains identified, and m refers to the sample weight (g or mL).

Please see lines 141–151. We hope that these changes/additions meet your expectation.

4. Results

-Table 1 should present the spore-pollen types for each cave separately, rather than in the current format. As it stands, it is difficult to determine which types are found in each cave. Given that the authors discuss both caves separately, as they should, I recommend restructuring the table to emphasise this distinction more clearly.

[Authors’ response] Your point is well taken. We have reorganized Table 1 (Lines 169–170) to clearly show the differences in pollen and spore types between the two caves.

-Correlation analysis: It is unclear which dataset was used for the correlation analysis. To ensure reproducibility and reusability, I recommend providing the dataset as a supplementary file (the specific pollen counts being compared should be explicitly stated for a meaningful interpretation). Furthermore, the statistical significance of the correlation (p-value) is not reported. To ensure a complete statistical interpretation, both R and p-values should be provided.

[Authors’ response] Sorry for the confusion. We have performed an updated correlation analysis, with full results (including p-values) now available in Supplementary s 1 and 2 Tables. In addition, we have incorporated significance level annotations in Figure 6 to enhance statistical clarity. Please see Line 221, Fig 6, S1 and S2 Tables. The dataset used for the correlation analysis has been organized in the Supporting Information. Please see Line 219, S1 and S2 Datasets.

5. Discussions and Conclusions

-Section: Cave environmental humidity – The relative humidity measured at each sampling point should be included in your assessment. As it stands, this section is general and primarily based on common knowledge that cave humidity is typically high. However, you do not provide any specific humidity measurements or discuss its variation in your results. Including this data would strengthen the analysis and help contextualise its potential influence on pollen preservation and deposition.

[Authors’ response] Thank you for your careful review. We agree. As one of the influence factors of surface sediment pollen assemblages in caves, environmental humidity may lead to significant spatial heterogeneity[1]. Quantitative measurement is one method to solve this problem. In this study, although we did not measurement cave humidity, previous research results was used to qualitatively indicate cave humidity[2-5]. Davis noted that dry cave environments, such as those found in Bechan in the western United States, are conducive to pollen preservation, while humid conditions tend to promote microbial activity, which accelerates pollen decay and makes the environment unfavorable for pollen preservation[2]. Thus, after careful consideration, we have added references 40 and 41 and modified the Discussion. Please see lines 302–308.

References:

1. Hunt CO, Fiacconi M. Pollen taphonomy of cave sediments: What does the pollen record in caves tell us about external environments and how do we assess its reliability ?. Quat Int. 2018; 485, 68–75.

2. Davis OK. Caves as sources of biotic remains in arid western North America. Palaeogeogr Palaeoclimatol Palaeoecol 1990; 76 (3–4): 331–348.

3. Navarro C, Carrión JS, Prieto AR, Munuera M. Modern cave pollen in an arid environment and its application to describe palaeorecords. Complutum. 2002; 13, 7–18.

4. Navarro C, Carrión J S, Munuera M, Prieto AR. Cave surface pollen and the palynological potential of karstic cave sediments in palaeoecology. Rev Palaeobot Palynol. 2001; 117, (4): 245‒265.

5. Navarro Camacho C, Carrión JS, Navarro J, Munuera M, Prieto AR. An experimental approach to the palynology of cave deposits. J Quat Sci. 2000; 15, (6): 603–619.

40. Yin C, Zhou Z, Cao M, Zhang J, Zhang Q, Zhang S, et al. Analysis on the relationship between climatic and environmental factors in karst caves: An example from Zhijin Cave of Guizhou Province. Carsologica sinica. 2016; 35 (4): 414–424. Chinese

41. Shi L, Zhou Z, Fan B, Tang Y, Yan L, An D, et al. Characteristics of Ventilation Effect in Karst Caves and Its Influence on Cave Air Environment: A Case Study of Mahuang Cave of Suiyang, Guizhou Province. Resour Environ Yangtze Basin. 2021; 30 (7): 1704–1713. Chinese

6. General comment:

The authors discuss factors influencing forest cover around the caves and its impact on pollen counts; however, these aspects are only vaguely mentioned, despite their crucial role in shaping the observed patterns. Forest cover can be easily inferred from existing maps using Land Use/Land Cover Classification (GIS & Spatial Analysis), a relatively simple and time-efficient approach that would enhance the study's interpretation. Alternatively, models estimating vegetation cover based on modern pollen deposition patterns in similar environments could provide valuable insights. Historical land cover data, aerial imagery, and satellite observations could also be used to compare present-day vegetation with past trends. Additionally, integrating environmental and climatic data, such as temperature, precipitation, and topographic features, would help contextualise vegetation distribution. To improve the robustness of the analysis, these factors should be incorporated into multifactorial analyses, either as quantitative or qualitative variables, to better assess the relationship between surface and subsurface pollen assemblages and their environmental drivers. If none of these approaches are feasible, the authors should explicitly acknowledge this limitation and discuss its potential impact on their conclusions.

[Authors’ response] Thank you for your comment. We agree. Land Use/Land Cover Classification (GIS & Spatial Analysis) was one of the easily methods to obtain the forest cover. However, its spatial resolution changed from 10 to 30 m. In this study, the area is very small. The impact of forest cover on the sedimentation inside the cave does not involve large-scale Land Use/Land Cover data.Thus, in this study, we conducted in-depth field investigations on the vegetation outside the cave and provided detailed descriptions in the revised version. Moreover, to incorporate your suggestion, we also add some background information in the “Description of the caves” section. Please see 2 Description of the caves. We hope that these changes/additions meet your expectation.

Minor comments and questions

1. Abstract

-Lines 14-15: The rationale for selecting these two specific caves remains unclear within the context provided by the authors.

[Authors’ response] Thank you for catching this. To address this, we have modified this sentence to: Cave pollen has great potential applications in reconstruction of past environments; However, the representativeness of cave pollen to the external environment remains controversial, especially in complex cave systems. This study conducted pollen analysis in surface sediment samples from two karst caves with complex geometry on the Guizhou Plateau. Please see lines 12–15.

2. Introduction

-Line 33: Providing a specific timespan would help clarify the temporal resolution of pollen analysis in reconstructing palaeovegetation or palaeoclimate.

[Authors’ response] Modified. Now this sentence changed to Fossil pollen preserved in these sediments has been used successfully in some case studies to reconstruct Quaternary paleovegetation and paleoclimate, and the method has shown great potential [6–13]. Please see lines 33–34.

3. Line 48: Could you please elaborate on what specific subsequent research you or the referenced authors are referring to ? Providing more context or examples would help clarify how this study contributes to or aligns with future research directions.

[Authors’ response] Sorry for the confusion. We have deleted this sentence after carefully consideration. Please see line 49.

4. Lines 74-76: The connection between the study’s aims and the archaeological records or past human occupation is unclear. As currently presented, these lines do not provide relevant context for the research focus and should be removed unless further justification is provided.

[Authors’ response] Thanks. We have removed it as suggested. Please see lines 74–76.

5. Lines 94-98, 112-115: Relevant references should be included to support these statements. If these descriptions are based on the authors’ own evaluations, they should be moved to the Results section rather than remaining in the introduction.

[Authors’ response] Thank you for catching this. We have added the reference in the text (Lines 92–96, Lines 111–114).

6. Results

-Line 202: The mention of surface sediment should be removed here, as it was already discussed in the previous paragraph. The current paragraph appears to focus on mosses, and keeping surface sediment in this context may cause confusion.

[Authors’ response] We apologize for the confusion caused by our previous wording. We have now corrected the term “surface sediment” to “surface soil.” Section 4.2 mainly presents the results of pollen analysis from both the inside and outside of the Dongkou Cave. The first paragraph describes the pollen analysis results from the cave interior’s surface sediment, while the second paragraph provides an overview of the pollen analysis results from the two types of surface samples collected outside the cave, namely surface soil sample and fresh moss sample. As we have already mentioned in the sample collection section, two different types of surface samples were collected outside the cave, including one surface soil sample and one fresh moss sample. Therefore, the second paragraph offers a comprehensive introduction to the pollen analysis results of these two samples. Please see line 210.

7. Discussion

-Line 225: Please clarify what the consistent pattern refers to or is in relation to. The statement lacks clarity and should specify the context or comparison being made.

[Authors’ response] Sorry for the confusion. The statement "consistent pattern" here refers to the similarity between the cave surface pollen assemblages and the external pollen rain. We have revised the relevant statements to make them clearer. Please see lines 233–236.

8. Lines 249-251: These statements belong to the Results section and should be moved accordingly for better structure and clarity.

[Authors’ response] Agree. We have integrated these statements into the Result section (Lines 219–227) as suggested.

9. Lines 274-276: On what grounds is this statement based ? Please provide more specificity, supporting data, or a relevant reference to substantiate the claim.

[Authors’ response] We apologize for the confusion. We have deleted this statement due to less supporting data. Please see line 283.

10. Lines 334-338: The phenomenon described, specific to caves modified for tourism or show caves, is known as "lampenflora". Please expand on this concept in your discussion and include relevant comparisons with existing literature that examines "lampenflora", its ecological implications, and how it relates to the findings of your study.

[Authors’ response] Thank you for the excellent suggestion. We have incorporated the concept of "lampenflora" into the revised version and expanded the discussion accordingly (Lines 346–350).

11. Lines 351-357: It would be valuable to elaborate further on how vegetation from different distances can be represented in pollen assemblages found inside caves. Are there estimates on the spatial scale of vegetation input based on existing studies? Vegetation maps detailing areas at varying proximities to the caves are likely available, and incorporating such data would enhance the study by providing an estimate of the pollen source area. This would be particularly useful for researchers working in this field, helping them contextualise similar findings

---

## [Decision Letter · Decision Letter 1]

27 May 2025

Dear Dr. Tang,

Thank you for submitting your manuscript to PLOS ONE. After careful consideration, we feel that it has merit but does not fully meet PLOS ONE’s publication criteria as it currently stands. Therefore, we invite you to submit a revised version of the manuscript that addresses the points raised during the review process.

We look forward to receiving your revised manuscript.

Kind regards,

Tzen-Yuh Chiang

Academic Editor

PLOS ONE

Reviewers' comments:

Reviewer's Responses to Questions

**Comments to the Author**

Reviewer #1: All comments have been addressed

Reviewer #2: (No Response)

Reviewer #3: (No Response)

2. Is the manuscript technically sound, and do the data support the conclusions?

Reviewer #1: Yes

Reviewer #2: Yes

Reviewer #3: Partly

3. Has the statistical analysis been performed appropriately and rigorously?

Reviewer #1: Yes

Reviewer #2: Yes

Reviewer #3: I Don't Know

4. Have the authors made all data underlying the findings in their manuscript fully available?

Reviewer #1: Yes

Reviewer #2: Yes

Reviewer #3: (No Response)

5. Is the manuscript presented in an intelligible fashion and written in standard English?

Reviewer #1: Yes

Reviewer #2: No

Reviewer #3: No

Reviewer #1: The authors have made a great effort to improve the manuscript and to address the issues raised during the review process. In my opinion, the work is now suitable for publication.

Reviewer #2: 1. There are some grammar errors in English expression in some places (such as adding 'a' before 'great' in the first sentence of the Abstract). It is recommended to have an English professional polish it;

2. In the Instruction, it is necessary to supplement the research on the paleoclimatic records of stalagmites, as it is an important geological historical archive material in karst areas;

3. The Line 32-34 statement indicates that since the paleovegetation and paleoclimate have been successfully reconstructed using pollen, the existing problems have been solved. Does this contradict the following problem description?

4. Suggest removing the reference "7" from Line 55, as it is a study from over 20 years ago and should not be classified as a "current research";

5. The starting point for the problem of multi-channel and multi cave is not the best. I suggest starting from scientific problems, that is, what kind of problems need to be solved through this study in the process of reconstructing paleoclimate and paleoenvironment using cave sediment pollen;

6. The description of the opening on Line 89-90 is inaccurate, with a width of 8m and a height of 4.5m. How could it be nearly circular?

7. What is the propagation distance of pollen in the research area, and what are the wind speed and direction? What is the scope of this research's investigation? These pieces of information need to be described clearly (Line 94-95);

8. Why is the complex cave feature repeatedly emphasized by the author not reflected in Line 234?

9. In Line 242, the specific distance may be related to the type and intensity of ventilation. How did the authors consider these factors?

10. Is a distance of tens of meters sufficient (Line 354)? What is the basis? How representative is it?

Reviewer #3: This is an interesting paper on cave pollen study. However, after I reviewed the revised manuscript, I do not think the authors took it seriously. The following concerns should be solved before publication.

1. There are several grammatical errors or typos. E.g.,

Line 39, ‘…but is also the key…’ should be ‘…, but also the key…’

Line 132, ‘using a DJI drone,)’, the comma after the word ‘drone’ should be deleted.

Line 144, ‘…after which the samples were again thoroughly washed again with…’ the first ‘again’ is a repeated word.

Line 233, ‘Our pollen analysis of complex caves with various geometry indicated that,, in both the multi-entrance’, the comma after the word ‘that’ should be deleted.

Line 236, ‘…pattern similar to that of the external pollen rain, albeit with minor differences (Fig 4–6).’ (Fig 4–6) should be ‘(Figs 4–6).’

2. There are three wrong Latin names in Table 1. Anacaodiaceae should be ‘Anacardiaceae’, Aluns should be ‘Alnus’, and Vsitis should be ‘Vitis’. Also Line 172, Aluns should be ‘Alnus’

3. I would like to see all the pollen and spore pictures. You said you found 69 taxa, then you should put all of them in the Fig.3

4. On the 4.3 Correlation analysis, I can not follow the numbers you stated. Lines 220-221, ‘…ranged from 0.18 to 0.98’, I cannot find the corresponding figures both in fig 6 and S1 and S2 tables. Plus, the significance levels are different. You use significance levels *P < 0.05, **P < 0.01, ***P < 0.001 in S1 and S2 Tables, but they are P<0.0001 in the table text.

**Do you want your identity to be public for this peer review?** For information about this choice, including consent withdrawal, please see our Privacy Policy

Reviewer #1: No

Reviewer #2: No

Reviewer #3: No

---

## [Author Response · Author response to Decision Letter 2]

20 Jun 2025

6. Review Comments to the Author

Reviewer #1:

The authors have made a great effort to improve the manuscript and to address the issues raised during the review process. In my opinion, the work is now suitable for publication.

[Authors’ response]

We sincerely appreciate your hard work and valuable comments on our manuscript; we are very grateful to hear that our revisions are acceptable.

Reviewer #2:

1. There are some grammar errors in English expression in some places (such as adding 'a' before 'great' in the first sentence of the Abstract). It is recommended to have an English professional polish it;

[Authors’ response]

Thank you for your careful review. We have implemented your suggestions (Line 12) and conducted a comprehensive revision of the manuscript; the editing certificate has been uploaded to the peer review system.

2. In the Instruction, it is necessary to supplement the research on the paleoclimatic records of stalagmites, as it is an important geological historical archive material in karst areas;

[Authors’ response]

Thank you for your careful review and constructive suggestions. Yes, stalagmite record is an important geological historical archive material in karst areas. This paper focuses on pollen assemblages and distribution characteristics in surface sediments of karst caves.

3. The Line 32-34 statement indicates that since the paleovegetation and paleoclimate have been successfully reconstructed using pollen, the existing problems have been solved. Does this contradict the following problem description?

[Authors’ response]

Thank you very much for pointing out the contradictory expressions in the later sections; We have made careful revisions accordingly. Please see Lines 33–34.

4. Suggest removing the reference "7" from Line 55, as it is a study from over 20 years ago and should not be classified as a "current research";

[Authors’ response]

Thank you. We have deleted the reference “7” in the text as suggested (Line 55).

5. The starting point for the problem of multi-channel and multi cave is not the best. I suggest starting from scientific problems, that is, what kind of problems need to be solved through this study in the process of reconstructing paleoclimate and paleoenvironment using cave sediment pollen;

[Authors’ response]

Thank you for your kind advise. In fact, current studies are mostly focused on simple caves with a single entrance and a single passage. However, research on more complex caves with multiple entrances, passages, and chambers remains very limited. Thus, the purpose of the study was to reveal the indicative role of complex cave surface pollen assemblages in reflecting external vegetation and to explore the potential factors influencing cave pollen taphonomic processes. Please see changes in “Abstract”.

6. The description of the opening on Line 89-90 is inaccurate, with a width of 8 m and a height of 4.5 m. How could it be nearly circular ?

[Authors’ response]

Thank you for catching this. We have corrected the cave entrance morphology to an elliptical shape. Please see Lines 86–87.

7. What is the propagation distance of pollen in the research area, and what are the wind speed and direction? What is the scope of this research's investigation ? These pieces of information need to be described clearly (Line 94-95);

[Authors’ response]

Thank you for your comments. In fact, the relationship between propagation distance of pollen and instantaneous wind is still challenging to study. However, palynologists have conducted studies on the relationship between modern cave pollen assemblages and external pollen rain and vegetation composition. In this study, we have modified Figures 1 and 2 to show the study scope in detail.

8. Why is the complex cave feature repeatedly emphasized by the author not reflected in Line 234?

[Authors’ response]

We apologize for the confusion. At the beginning of section 5.1, we have already reaffirmed the structure of the two complex cave systems, as shown in Lines 232–234—Our pollen analysis of complex caves with various geometry indicated that in both the multi-entrance multi-passage Yinhegong Cave and the single-entrance multi-chamber Dongkou Cave with significant variations in passage width, ….

9. In Line 242, the specific distance may be related to the type and intensity of ventilation. How did the authors consider these factors?

[Authors’ response]

Thank you for your careful comments. In fact, the relationship between propagation distance of pollen and instantaneous wind is still challenging to study. In complex caves, instantaneous wind speed changes constantly. In the present study, in both the multi-entrance, multi-passage Yinhegong Cave and the single-entrance, multi-chamber Dongkou Cave, with significant passage width variation, pollen assemblage in the surface sediment within 5–15 m of the entrance exhibited strong correspondence with external pollen rain, effectively reflecting the composition of the external vegetation. Our study can only reflect the existence of this phenomenon and new finds. The formation mechanism needs to be analyzed further with long-term monitoring.

10. Is a distance of tens of meters sufficient (Line 354)? What is the basis? How representative is it?

[Authors’ response]

Thank you. We designed this study to collect surface soil and fresh moss samples within tens of meters around the cave entrance for modern pollen rain analysis. This experimental approach was adopted because it remains unclear whether pollen assemblages in cave surface sediments correspond to external vegetation composition. Even if such correspondence exists, the spatial range over which these pollen assemblages can accurately reflect external vegetation characteristics is also uncertain in our study area. Thus, this experiment serves as an exploratory investigation to address these uncertainties.

The rationale for our experimental design is twofold. First, it is based on the definitions of local and regional spatial scales outside the cave provided by Coles et al. (1898). Second, we drew on the spatial distribution ranges of modern sample collection points used in previous studies, such as those by Navarro et al. (2001, 2002), de Porras et al. (2011), Fiacconi and Hunt (2017), and Yang et al. (2021). The studies typically confined their sampling to within a 200-m radius around the cave entrance. Our decision to focus on a smaller radius (tens of meters) around the cave entrance is intended to provide a more detailed and localized understanding of the relationship between cave pollen assemblages and external vegetation.

Our study reveals that the surface pollen assemblages within the 5–15 m zone at the front of the cave entrance exhibit good correspondence with the external vegetation. This finding is consistent with the research of de Porras et al. (2011), Fiacconi and Hunt (2015), and Yang et al. (2021) However, the relationship between the pollen assemblages and vegetation at greater spatial distances remains unclear. We have explicitly acknowledged this limitation in our discussion (Lines 366–371) and suggest that future research should address this gap by increasing the collection of modern samples and conducting vegetation surveys across different spatial ranges outside the cave. This approach will help to better define the spatial representativeness of cave surface pollen assemblages.

References:

Coles GM, Gilbertson DD, Hunt CO, Jenkinson RDS. Taphonomy and the palynology of cave deposits. Cave Sci. 1989; 16 (3): 83–89.

Navarro C, Carrión JS, Munuera M, Prieto AR. Cave surface pollen and the palynological potential of karstic cave sediments in palaeoecology. Rev Palaeobot Palynol. 2001; 117, (4): 245‒265.

Navarro C, Carrión JS, Prieto AR, Munuera M. Modern cave pollen in an arid environment and its application to describe palaeorecords. Complutum. 2002; 13, 7–18.

de Porras ME, Mancini MV, Prieto AR. Modern pollen analysis in caves at the Patagonian steppe, Argentina. Rev Palaeobot Palynol. 2011; 166, 335‒343.

Fiacconi M, Hunt CO. Pollen taphonomy at Shanidar Cave (Kurdish Iraq): An initial evaluation. Rev Palaeobot Palynol. 2015; 223, 87–93.

Fiacconi M, Hunt CO. Palynology of surface sediments from caves in the Zagros Mountains (Kurdish Iraq): Patterns and processes. Rev Palaeobot Palynol. 2017; 239, 66–76.

Yang Q, Zhao K, Zhou X, Wang J, Chen G, Li D, et al. Evaluation of the potential of surface pollen spectra from caves in SW China for vegetation reconstruction. Quat Int. 2021; 591, 119–128.

Reviewer #3:

This is an interesting paper on cave pollen study. However, after I reviewed the revised manuscript, I do not think the authors took it seriously. The following concerns should be solved before publication.

1. There are several grammatical errors or typos. E.g.,

Line 39, ‘…but is also the key…’ should be ‘…, but also the key…’

Line 132, ‘using a DJI drone,)’, the comma after the word ‘drone’ should be deleted.

Line 144, ‘…after which the samples were again thoroughly washed again with…’ the first ‘again’ is a repeated word.

Line 233, ‘Our pollen analysis of complex caves with various geometry indicated that,, in both the multi-entrance’, the comma after the word ‘that’ should be deleted.

Line 236, ‘…pattern similar to that of the external pollen rain, albeit with minor differences (Fig 4–6).’ (Fig 4–6) should be ‘(Figs 4–6).’

[Authors’ response]

Thank you for your careful review. We have corrected the above grammatical errors and typos according to your suggestions.

2. There are three wrong Latin names in Table 1. Anacaodiaceae should be ‘Anacardiaceae’, Aluns should be ‘Alnus’, and Vsitis should be ‘Vitis’. Also Line 172, Aluns should be ‘Alnus’

[Authors’ response]

Thank you for catching these. We have corrected the wrong Latin names in Table 1, Figures 4 and 5, and S1 and S2 Datasets, and updated the figures.

3. I would like to see all the pollen and spore pictures. You said you found 69 taxa, then you should put all of them in the Fig.3

[Authors’ response]

Thank you very much for your suggestion. Since the focus of our study is not on pollen morphology, we did not take photos of all the pollen types during the identification. Consequently, we apologize that we cannot provide photos of the 69 pollen taxa mentioned in this paper. To avoid inconsistency between the text and the figure, we have revised the caption of Figure 3 accordingly (Line 170).

4. On the 4.3 Correlation analysis, I can not follow the numbers you stated. Lines 220-221, ‘…ranged from 0.18 to 0.98’, I cannot find the corresponding figures both in fig 6 and S1 and S2 tables. Plus, the significance levels are different. You use significance levels *P < 0.05, **P < 0.01, ***P < 0.001 in S1 and S2 Tables, but they are P<0.0001 in the table text.

[Authors’ response]

We apologize for the confusion. Thank you very much for pointing out this issue. We have carefully checked the original results and found that the data in Figure 6 were misaligned. Additionally, the R values in SI and S2 Tables had not been rounded off, whereas the R values shown in Figure 6 had indeed been rounded off. Therefore, we have revised and updated the Figure 6 and SI and S2 Tables to ensure consistency in the results.

The range of the correlation coefficients (R) has been revised to 0.18–0.99, with the specific results highlighted using yellow and red rectangular boxes in Figure 6.

Fig 6. Correlation coefficients for pollen spectra from the samples of Yinhegong Cave (A) and Dongkou (B) Cave (significance levels *: P < 0.05; **: P < 0.01; ***: P < 0.001)

In SI and S2 Tables, the values in rows labeled "P" (for example, the values highlighted with a yellow background in SI Table) represent the specific P-values at different significance levels (significance levels P < 0.05, P < 0.01, and P < 0.001); that is, the probability of the observed sample data occurring. To avoid confusion, we have revised the labels "P" to "P-value" in SI and S2 Tables.

S1 Table. Correlation coefficients for the pollen spectra from the samples at Yinhegong Cave (significance levels * P < 0.05, ** P < 0.01, *** P < 0.001)

Ym Y05 Y10 Y15 Y25 Y30

Ym R 1.000

P —

Y05 R 0.96*** 1.000

P <0.0001 —

Y10 R 0.95*** 1.00*** 1.000

P <0.0001 <0.0001 —

Y15 R 0.66*** 0.83*** 0.85*** 1.000

P <0.0001 <0.0001 <0.0001 —

Y25 R 0.94*** 0.99*** 0.99*** 0.87*** 1.000

P <0.0001 <0.0001 <0.0001 <0.0001 —

Y30 R 0.27 0.49** 0.51*** 0.86*** 0.56*** 1.000

P 0.118 0.002 0.001 <0.0001 0.000 —

S1 and S2 Tables have been revised to the following format.

Thank you for your comment again. Your comments are valid and challenging. To incorporate your points, we also checked our manuscript carefully. We believe the changes/additions made have improved the quality of the paper substantially, and we hope that they meet your expectations.

---

## [Decision Letter · Decision Letter 2]

15 Jul 2025

Dear Dr. Tang,

Thank you for submitting your manuscript to PLOS ONE. After careful consideration, we feel that it has merit but does not fully meet PLOS ONE’s publication criteria as it currently stands. Therefore, we invite you to submit a revised version of the manuscript that addresses the points raised during the review process.

We look forward to receiving your revised manuscript.

Kind regards,

Tzen-Yuh Chiang

Academic Editor

PLOS ONE

Journal Requirements:

Reviewers' comments:

Reviewer's Responses to Questions

**Comments to the Author**

Reviewer #1: All comments have been addressed

Reviewer #2: All comments have been addressed

2. Is the manuscript technically sound, and do the data support the conclusions?

Reviewer #1: Yes

Reviewer #2: Partly

3. Has the statistical analysis been performed appropriately and rigorously?

Reviewer #1: Yes

Reviewer #2: N/A

4. Have the authors made all data underlying the findings in their manuscript fully available?

Reviewer #1: Yes

Reviewer #2: Yes

5. Is the manuscript presented in an intelligible fashion and written in standard English?

Reviewer #1: Yes

Reviewer #2: No

Reviewer #1: The authors have carefully considered the reviewers’ comments and addressed them where appropriate, while remaining aligned with the original scope and objectives of their study. The revisions have improved the clarity and presentation of the manuscript. In my view, the work is now suitable for publication.

Reviewer #2: This manuscript has passed two rounds of expert reviews and revisions, but I believe that overall there has been no substantial improvement, especially in answering most of the questions raised by the reviewers, which were not fully adopted in the manuscript revision and did not provide convincing reasons for rejecting the revisions. Include discussions on paleoclimate records such as stalagmites and scientific questions (research objectives) in the introduction section. And many contradictory statements still exist in many places. In addition, there is no necessary description of the cave sediments mentioned in the manuscript, and it is difficult to determine whether they are suitable as archives for paleoclimate reconstruction (at least dating and sedimentary sequences must be regular). Therefore, I have to suggest rejection.

**Do you want your identity to be public for this peer review?** For information about this choice, including consent withdrawal, please see our Privacy Policy

Reviewer #1: No

Reviewer #2: No

---

## [Author Response · Author response to Decision Letter 3]

21 Jul 2025

Reviewers' comments:

6. Review Comments to the Author

Reviewer #1: The authors have carefully considered the reviewers’ comments and addressed them where appropriate, while remaining aligned with the original scope and objectives of their study. The revisions have improved the clarity and presentation of the manuscript. In my view, the work is now suitable for publication.

[Authors’ response]

We sincerely appreciate your constructive feedback and are pleased that our revisions have addressed your concerns effectively. Your insightful suggestions significantly improved the manuscript’s clarity and scholarly rigor. We are grateful for your time and expertise in evaluating our work and are honored by your endorsement for publication.

Reviewer #2: This manuscript has passed two rounds of expert reviews and revisions, but I believe that overall there has been no substantial improvement, especially in answering most of the questions raised by the reviewers, which were not fully adopted in the manuscript revision and did not provide convincing reasons for rejecting the revisions. Include discussions on paleoclimate records such as stalagmites and scientific questions (research objectives) in the introduction section. And many contradictory statements still exist in many places. In addition, there is no necessary description of the cave sediments mentioned in the manuscript, and it is difficult to determine whether they are suitable as archives for paleoclimate reconstruction (at least dating and sedimentary sequences must be regular). Therefore, I have to suggest rejection.

[Authors’ response]

We sincerely thank you for your constructive comments. Nevertheless, we regret that our previous revision did not fully meet your expectations, and we are truly sorry for this shortcoming. Consequently, we have carefully re-examined the two rounds of feedback provided by all three reviewers, paying particular attention to the issues you raised in both your previous and the present comments. On this basis, we have undertaken another thorough revision, hoping that the changes will further improve the quality of the manuscript.

Below, we address the three specific points raised in your latest comments and detail the corresponding modifications and clarifications.

(1) Include discussions on paleoclimate records such as stalagmites and scientific questions (research objectives) in the introduction section.

[Authors’ response]

We sincerely thank you for your careful review and constructive comments. While we fully acknowledge that speleothems—particularly stalagmites—offer outstanding advantages in palaeoclimate reconstruction because of their high-resolution geochemical archives, this paper is explicitly devoted to the relationship between modern cave pollen assemblages and the external vegetation and environment. This modern process investigation is designed to serve as an analogue for clarifying how fossil pollen preserved in cave sediments reflects contemporaneous external conditions.

Accordingly, we have refrained from providing an in-depth review or critical assessment of stalagmite-based palaeoclimate research in the Introduction. Instead, we focus on clearly articulating the objectives and significance of the modern cave pollen study (Please see Lines: 36–43). Furthermore, a survey of the international literature specifically focused on modern cave pollen analysis (e.g., Fiacconi and Hunt, 2015, 2017; Hunt and Fiacconi, 2018; Navarro et al., 2001, 2012; Yang et al., 2021) reveals that these papers seldom introduces stalagmites or their palaeoclimate applications in either the Introduction or the Discussion. For these reasons, we believe that a discussion of stalagmite-based palaeoclimate research is not essential to this paper, and we therefore prefer not to add such content.

In this revised version, we have further clarified the scientific questions and refined the corresponding statements— ... whether fossil pollen preserved in cave sediments faithfully reflects contemporaneous environmental information beyond the cave [5, 14–18]. To reduce these uncertainties, research into modern cave pollen processes is indispensable, as it provides the critical modern analogue for interpreting the taphonomy and environmental significance of such archives [14, 19, 20]—in the Introduction. Please see Lines 36–40.

References:

Fiacconi M, Hunt CO. Pollen taphonomy at Shanidar Cave (Kurdish Iraq): An initial evaluation. Rev Palaeobot Palynol. 2015; 223, 87–93.

Fiacconi M, Hunt CO. Palynology of surface sediments from caves in the Zagros Mountains (Kurdish Iraq): Patterns and processes. Rev Palaeobot Palynol. 2017; 239, 66–76.

Hunt CO, Fiacconi M. Pollen taphonomy of cave sediments: What does the pollen record in caves tell us about external environments and how do we assess its reliability ?. Quat Int. 2018; 485, 68–75.

Navarro C, Carrión JS, Munuera M, Prieto AR. Cave surface pollen and the palynological potential of karstic cave sediments in palaeoecology. Rev Palaeobot Palynol. 2001; 117, (4): 245‒265.

Navarro C, Carrión JS, Prieto AR, Munuera M. Modern cave pollen in an arid environment and its application to describe palaeorecords. Complutum. 2002; 13, 7–18.

Yang Q, Zhao K, Zhou X, Wang J, Chen G, Li D, et al. Evaluation of the potential of surface pollen spectra from caves in SW China for vegetation reconstruction. Quat Int. 2021; 591, 119–128.

(2) And many contradictory statements still exist in many places.

[Authors’ response]

Thank you for your meticulous and responsible review, which brought to light several contradictory statements in our manuscript. We have thoroughly re-examined the text and have made the following revisions accordingly.

We have revised “Fern spores were mainly represented by Gleicheniaceae (5.0%) and Polypodiaceae (3.5%), with contents decreasing toward the interior of the cave (Fig 5B)” to “Fern spores were mainly represented by Gleicheniaceae (5.0%) and Polypodiaceae (3.5%), showing higher percentages in both the entrance and rear sections (Fig 5B)”. (Lines 203–204)

The contradictory statement noted in the previous round review—Fossil pollen preserved in these sediments has been used successfully in some case studies to reconstruct Quaternary paleovegetation and paleoclimate, and the method has shown great potential—was removed during the last revision and has been replaced with the following wording: Fossil pollen sequences preserved in these sediments hold great potential for reconstructing Quaternary paleovegetation and paleoclimate changes [6–13]. (Lines 31–34)

(3) In addition, there is no necessary description of the cave sediments mentioned in the manuscript, and it is difficult to determine whether they are suitable as archives for paleoclimate reconstruction (at least dating and sedimentary sequences must be regular).

[Authors’ response]

We sincerely thank you for your professional suggestions from the perspectives of sedimentology and geochronology. Indeed, the establishment of a high-precision, reliable chronological framework and the detailed description of stratigraphic sequences—including lithology and stratigraphic contact—constitute the indispensable foundation for any robust paleoenvironmental reconstruction.

This study focuses primarily on the relationship between modern cave pollen and external vegetation—in other words, on modern pollen processes within the cave—rather than on paleoclimate reconstruction based on cave sedimentary strata. Consequently, the samples collected here consist only of the uppermost ~1 cm surface layer on the cave floor (see Materials and Methods, lines 117–118), not a deep stratigraphic profile. Therefore, chronological dating and descriptions of lithology and stratigraphic contacts are not required.

In addition, we have corrected the spelling errors in Figures 4 and 5 and updated the figures accordingly. For all other detailed revisions, please refer to “Revised Manuscript with Track Changes”.

Thank you for your comments again. Your comments are valid and challenging. To incorporate your points, we also checked our manuscript carefully. We believe the changes/additions made have improved the quality of the paper substantially, and we hope that they meet your expectations.

---

## [Decision Letter · Decision Letter 3]

13 Aug 2025

Dear Dr. Tang,

Thank you for submitting your manuscript to PLOS ONE. After careful consideration, we feel that it has merit but does not fully meet PLOS ONE’s publication criteria as it currently stands. Therefore, we invite you to submit a revised version of the manuscript that addresses the points raised during the review process.

We look forward to receiving your revised manuscript.

Kind regards,

Tzen-Yuh Chiang

Academic Editor

PLOS ONE

Journal Requirements:

Reviewers' comments:

Reviewer's Responses to Questions

**Comments to the Author**

Reviewer #1: All comments have been addressed

Reviewer #4: All comments have been addressed

Reviewer #5: All comments have been addressed

2. Is the manuscript technically sound, and do the data support the conclusions?

Reviewer #1: Yes

Reviewer #4: Yes

Reviewer #5: Yes

3. Has the statistical analysis been performed appropriately and rigorously?

Reviewer #1: Yes

Reviewer #4: Yes

Reviewer #5: Yes

4. Have the authors made all data underlying the findings in their manuscript fully available?

Reviewer #1: Yes

Reviewer #4: Yes

Reviewer #5: Yes

5. Is the manuscript presented in an intelligible fashion and written in standard English?

Reviewer #1: Yes

Reviewer #4: Yes

Reviewer #5: Yes

Reviewer #1: I stand by my decision that the authors have carefully considered the reviewers’ comments and addressed them where appropriate, while remaining aligned with the original scope and objectives of their study. The revisions have improved the clarity and overall presentation of the manuscript.

In my view, the work is suitable for publication. The authors have remained focused on the topic as specified in the title and, as agreed in the previous revision, have removed sections not directly related to it. They have strengthened the relevant sections using appropriate methodology, clearly presented the results, and discussed them in their proper context.

Reviewer #4: Although cave sediment is not ideal for reconstructing past environments and climate, it holds significant importance for studying ancient humans and their habitats. I recommend the authors refine their presentation of this point in the Abstract and Introduction.

Reviewer #5: The manuscript reports an interesting and important study on surface pollen in karst caves. Cave sediments in some regions, especially regions without good and long-enough lake sediments or continuous aeolian deposits, are good records for paleoenvironment reconstruction. I am glad to see the efforts paid by these authors trying to understand the basics of pollen studies in karst caves. A careful evaluation of pollen deposited in and outside the cave is the premise to do paleo-pollen studies, which not only important for past climate change studies, but also for archaeological studies in caves where archaeological remains preserved. After three round of review, I saw that previous reviewers had provided good suggestions and the authors had dealt with them carefully. And the manuscript has been improved a lot since the original one. I think the current manuscript is almost ready for publication. But please still try to consider my following suggestion:

I understand that the authors had thought about the influence of the structure of caves on the transportation and deposition of pollen. In fact, it is one of the reason why the authors did this study in these two caves. I would suggest to provide a map of the vertical profile of the sampled part of the cave to show the slope of the cave surface. It is important for the transportation of pollen and sediments in the cave, which also effects the explanation of the pollen results in this study. Therefore, the influence of geomorphology of the inside of the cave on the pollen results should be carefully considered.

**Do you want your identity to be public for this peer review?** For information about this choice, including consent withdrawal, please see our Privacy Policy

Reviewer #1: No

Reviewer #4: No

Reviewer #5: No

---

## [Author Response · Author response to Decision Letter 4]

15 Aug 2025

Reviewers' comments:

6. Review Comments to the Author

Reviewer #1: I stand by my decision that the authors have carefully considered the reviewers’ comments and addressed them where appropriate, while remaining aligned with the original scope and objectives of their study. The revisions have improved the clarity and overall presentation of the manuscript.

In my view, the work is suitable for publication. The authors have remained focused on the topic as specified in the title and, as agreed in the previous revision, have removed sections not directly related to it. They have strengthened the relevant sections using appropriate methodology, clearly presented the results, and discussed them in their proper context.

[Authors’ response]

Thank you very much for your thorough review and for confirming that our revisions have adequately addressed all comments while remaining faithful to the original scope of the study. We appreciate your positive assessment of the improved clarity and presentation, as well as your endorsement of the manuscript’s suitability for publication. Your guidance has been invaluable in helping us strengthen the work, and we are grateful for your continued support.

Reviewer #4: Although cave sediment is not ideal for reconstructing past environments and climate, it holds significant importance for studying ancient humans and their habitats. I recommend the authors refine their presentation of this point in the Abstract and Introduction.

[Authors’ response]

We are truly grateful for your constructive feedback. We agree that cave sediments are irreplaceable for understanding ancient humans and their habitats. In response, we have revised the Abstract to foreground the archaeological significance of our record (lines 12–14). The Introduction already contains an extended discussion of the same issue (Please see Lines 62–71); no further changes were necessary there.

Reviewer #5: The manuscript reports an interesting and important study on surface pollen in karst caves. Cave sediments in some regions, especially regions without good and long-enough lake sediments or continuous aeolian deposits, are good records for paleoenvironment reconstruction. I am glad to see the efforts paid by these authors trying to understand the basics of pollen studies in karst caves. A careful evaluation of pollen deposited in and outside the cave is the premise to do paleo-pollen studies, which not only important for past climate change studies, but also for archaeological studies in caves where archaeological remains preserved. After three round of review, I saw that previous reviewers had provided good suggestions and the authors had dealt with them carefully. And the manuscript has been improved a lot since the original one. I think the current manuscript is almost ready for publication. But please still try to consider my following suggestion:

I understand that the authors had thought about the influence of the structure of caves on the transportation and deposition of pollen. In fact, it is one of the reason why the authors did this study in these two caves. I would suggest to provide a map of the vertical profile of the sampled part of the cave to show the slope of the cave surface. It is important for the transportation of pollen and sediments in the cave, which also effects the explanation of the pollen results in this study. Therefore, the influence of geomorphology of the inside of the cave on the pollen results should be carefully considered.

[Authors’ response]

Thank you very much for your insightful comments. In response to your suggestion, we have incorporated field-measured cave-morphology data to generate vertical-profile maps of both Yinhegong Cave and Dongkou Cave, which are now presented as part of Figure 2.

The updated Figure 2 is provided below.

Once more, we thank you for your incisive remarks. Your observations were both pertinent and thought-provoking. After a thorough re-examination of the manuscript, we have integrated these suggestions and are confident that the resulting revisions have markedly strengthened the paper; we hope that these changes now satisfy your expectations.

---

## [Decision Letter · Decision Letter 4]

8 Sep 2025

Pollen assemblages and distribution characteristics in surface sediments of karst caves on the Guizhou Plateau, southwestern China

PONE-D-25-10253R4

Dear Dr. Tang,

We’re pleased to inform you that your manuscript has been judged scientifically suitable for publication and will be formally accepted for publication once it meets all outstanding technical requirements.

Kind regards,

Tzen-Yuh Chiang

Academic Editor

PLOS ONE

Additional Editor Comments (optional):

Reviewer #1:

Reviewer #4:

Reviewer #5:

Reviewers' comments:

Reviewer's Responses to Questions

**Comments to the Author**

Reviewer #1: All comments have been addressed

Reviewer #4: All comments have been addressed

Reviewer #5: All comments have been addressed

2. Is the manuscript technically sound, and do the data support the conclusions?

Reviewer #1: Yes

Reviewer #4: Yes

Reviewer #5: Yes

3. Has the statistical analysis been performed appropriately and rigorously?

Reviewer #1: Yes

Reviewer #4: Yes

Reviewer #5: Yes

4. Have the authors made all data underlying the findings in their manuscript fully available?

Reviewer #1: Yes

Reviewer #4: Yes

Reviewer #5: Yes

5. Is the manuscript presented in an intelligible fashion and written in standard English?

Reviewer #1: Yes

Reviewer #4: Yes

Reviewer #5: Yes

Reviewer #1: The authors have addressed all reviewer comments to the best of their ability, aiming to incorporate the requested revisions while maintaining the manuscript’s original perspective and scientific interpretation.

Reviewer #4: (No Response)

Reviewer #5: I am satisfy to see the revise figure 2 with the vertical profile map of the sampled part of the two caves. I agree to accept the paper to publish at this stage.

**Do you want your identity to be public for this peer review?** For information about this choice, including consent withdrawal, please see our Privacy Policy

Reviewer #1: No

Reviewer #4: No

Reviewer #5: No

---

## [Editor Report · Acceptance letter]

PONE-D-25-10253R4

PLOS ONE

Dear Dr. Tang,

I'm pleased to inform you that your manuscript has been deemed suitable for publication in PLOS ONE. Congratulations! Your manuscript is now being handed over to our production team.

Kind regards,

on behalf of

Dr. Tzen-Yuh Chiang

Academic Editor

PLOS ONE